# Act Only When It Pays: Efficient Reinforcement Learning for LLM Reasoning via Selective Rollouts

**Haizhong Zheng**[†]  **Yang Zhou**[†]  **Brian R. Bartoldson**[‡]

**Bhavya Kailkhura**[‡]  **Fan Lai**[§]  **Jiawei Zhao**[¶]  **Beidi Chen**[†]

[†]Carnegie Mellon University  [‡]Lawrence Livermore National Laboratory
[§]University of Illinois Urbana-Champaign  [¶]Meta AI (FAIR)

## Abstract

Reinforcement learning, such as PPO and GRPO, has powered recent breakthroughs in LLM reasoning. Scaling rollout to sample more prompts enables models to selectively use higher-quality data for training, which can stabilize RL training and improve model performance, but at the cost of significant computational overhead. In this paper, we first show that a substantial portion of this overhead can be avoided by skipping uninformative prompts *before rollout*. Our analysis of reward dynamics reveals a strong temporal consistency in prompt value: prompts that are uninformative in one epoch of training are likely to remain uninformative in near future epochs. Based on these insights, we propose GRESO (GRPO with Efficient Selective Rollout), an online, lightweight pre-rollout filtering algorithm that predicts and skips uninformative prompts using reward training dynamics. By evaluating GRESO on a broad range of math reasoning benchmarks and models, like Qwen2.5-Math-1.5B, DeepSeek-R1-Distill-Qwen-1.5B, Qwen2.5-Math-7B, Qwen2.5-14B, and Qwen2.5-32B, we show that GRESO achieves up to **2.4×** **wall-clock time speedup** in rollout and up to **2.0× speedup** in total training time without accuracy degradation. We make our code publicly available at GitHub[1].

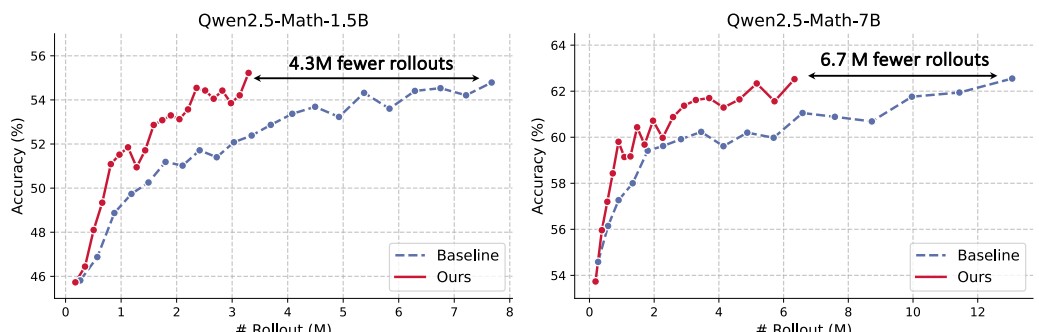

Figure 1: We train Qwen2.5-Math-1.5B/7B on the DAPO + MATH dataset and evaluate them on five math reasoning benchmarks: MATH500, AMC, Gaokao, Minerva, and Olympiad Bench. Compared to the baseline method (Dynamic Sampling), our approach (GRESO) reduces rollout overhead by up to 2× while achieving comparable training performance, improving the efficiency of rollout scaling.

---

[1]https://github.com/Infini-AI-Lab/GRESO/

39th Conference on Neural Information Processing Systems (NeurIPS 2025).

# 1  Introduction

Recent reasoning models [9, 34, 40], such as OpenAI's o1 and DeepSeek's R1, leverage Chain-of-Thought as a form of test-time scaling to significantly enhance the reasoning capabilities of large language models (LLMs). Reinforcement Learning (RL) techniques, including PPO [35] and GRPO [9], have emerged as key drivers of this progress. By generating data online during each training iteration (i.e., rollout), reinforcement learning enables models to iteratively refine their reasoning strategies through self-exploration, often achieving or even surpassing human-level performance [34, 38, 39]. Notably, *scaling computational resources to sample responses for more prompts* at this rollout stage can further enhance training, which allows models to selectively utilize higher-quality data and thus train models with better converged performance [46, 49]. However, scaling up rollouts introduces significant computational overhead, as rollout remains a major bottleneck in RL training [33, 37, 42, 58]. For instance, as shown in Figure 2, filtering out uninformative examples[2] and resampling to fill the batch with effective data (also known as Dynamic Sampling in [49]) can improve model performance, but it comes at the cost of significantly increased rollout overhead. Motivated by this challenge, we aim to investigate the following research question:

> *How can we perform more selective rollouts—focusing on sampling more valuable prompts—to make this rollout scaling more efficient?*

Existing methods face several limitations in addressing this question. First, some approaches [24, 43] attempt to improve data efficiency by pruning datasets before training. These methods typically rely on training a model to identify valuable data points; however, there is no conclusive evidence that such strategies improve the overall efficiency of RL training as well. Second, these static pruning methods overlook the fact that the value of a data point can vary across models and training stages, limiting their ability to support adaptive data selection. Finally, online selection approaches such as Dynamic Sampling [49] perform oversampling and filter out uninformative data only after rollout, leading to substantial additional rollout cost. Estimating data quality accurately and efficiently *before rollout* remains a challenging and underexplored problem.

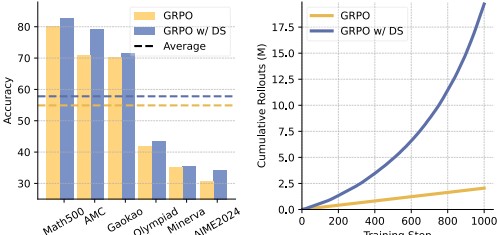

Figure 2: **Left:** GRPO training with more effective data through Dynamic Sampling (DS) leads to improved final model performance. **Right:** However, DS requires additional rollouts to maintain the same training batch size.

Consequently, an ideal selective rollout algorithm for efficient LLM RL should have the following properties: **1) Online data selection.** Instead of relying on an auxiliary model trained offline to pre-prune the dataset, an ideal method should perform data selection online during training. This avoids the additional overhead of training a separate model and enables decisions to be made based on the current training states. **2) Model-based data value estimation.** Data values evolve throughout training and vary across different models, requiring a selective rollout strategy to adapt dynamically to different models and training stages. **3) Low computational overhead.** To ensure scalability, the selective rollout strategy should introduce minimal additional cost during training.

In this paper, we aim to design an efficient selective rollout strategy for LLM RL to make rollout scaling more efficient. We begin by analyzing the training dynamics of prompts across epochs and observe a strong temporal consistency across different training epochs (Section 3). In particular, prompts that yield zero advantage for all sampled responses in one epoch are more likely to do so in future epochs as well. This temporal correlation suggests that historical reward dynamics can be leveraged to predict and preemptively skip those uninformative prompts before rollout. Building on these observations, we propose **GRESO** (G̲RPO with E̲fficient S̲elective R̲ollout) in Section 4, depicted in Figure 4b, an online efficient pre-rollout filtering algorithm that reduces rollout cost by selectively skipping prompts predicted to be uninformative. Instead of performing filtering after rollout, GRESO estimates a skipping probability for each prompt based on its reward dynamics

---

[2]In GRPO, many examples yield identical rewards across all responses, resulting in zero advantage and thus contributing no learning signal during training.

during training prior to the rollout stage, significantly reducing prompt selection overhead and making the rollout scaling more efficient.

In Section 5, we empirically verify the efficiency of GRESO on six math reasoning benchmarks and five models: Qwen2.5-Math-1.5B [47], DeepSeek-R1-Distill-Qwen-1.5B [9], Qwen2.5-Math-7B [47], Qwen2.5-14B [47], and Qwen2.5-32B [47]. Our evaluation results show that GRESO achieves up to **2.4× speedup** in rollout and **2.0× speedup** in total training time while maintaining comparable accuracy (Section 5.2). We also conduct a more detailed study on how GRESO reduces training overhead by performing selective rollout and ablation study on different components of GRESO in Section 5.3.

## 2   Related Work

**RL for LLM Reasoning.** Reinforcement learning (RL) was initially used to align model outputs with human preferences [7, 35]. Since then, RL has become a commonly used technique for fine-tuning LLMs, enabling them to generate more helpful, harmless, and honest responses by incorporating reward signals from human feedback [3, 6]. Recent advances [9, 12, 40, 49] in LLM reasoning show that Reinforcement Learning with Verifiable Reward (RLVR), which relies on verifiable reward signals instead of model-generated scoress, can effectively improve model reasoning ability. These gains are achieved using various policy optimization methods such as PPO [35] and GRPO [36]. Encouraged by the success of RLVR, a growing body of work [15, 19, 27, 30, 45, 49–52] has emerged to further improve reinforcement learning methods for LLM reasoning. For instance, methods such as VinePPO [19], VC-PPO [51], and VAPO [50] aim to enhance LLM reasoning by optimizing the value function Meanwhile, DAPO [49] introduces several techniques to improve GRPO, including Dynamic Sampling, which filters out zero-variance prompts and refills the training batch with effective training data through resampling.

**Data Selection for LLM**. In addition to improving training algorithms, another line of work [17, 31, 44, 48] seeks to enhance the efficiency and effectiveness of LLM training through data selection strategies. Several approaches [5, 16, 44] focus on pruning data used for supervised fine-tuning. For example, S1 [31] reduces a large set of 59k examples to just 1k high-quality samples. In parallel, another thread of research [8, 11, 24, 28, 32, 43] targets improving data efficiency in reinforcement learning for LLMs. For instance, recent research [24, 43] shows that only a small subset of the original training dataset is necessary for GRPO to improve the model's reasoning ability. However, those methods rely on training models with the full dataset first to identify important samples and do not offer clear improvements in end-to-end RL training efficiency.

## 3   Observation

In this section, we study the impact of uninformative prompts—specifically, zero-variance prompts—on GRPO training. We empirically show that a high zero-variance ratio can hurt the training performance (Section 3.2). Our analysis reveals a strong temporal consistency in prompt value: prompts that are uninformative in one training epoch tend to remain uninformative in future epochs, which inspires the design of GRESO (Section 3.3).

### 3.1   Background: Group Relative Policy Optimization (GRPO)

Group Relative Policy Optimization (GRPO)[36] is a variant of Proximal Policy Optimization (PPO) [35] tailored for language model fine-tuning. Instead of computing advantages using a value function, GRPO normalizes reward scores within groups of responses sampled for the same prompt, which largely improves the training efficiency. GRPO has shown superior performance in recent advances [9, 24, 43, 49] in RL for LLMs, especially for reasoning tasks. GRPO aims to maximize the following objective:

$$\mathcal{J}_{GRPO}(\theta) = \mathbb{E}[q \sim P(Q), \{o_i\}_{i=1}^G \sim \pi_{\theta_{old}}(O|q)]$$
$$\frac{1}{G}\sum_{i=1}^{G}\left(\min\left(\frac{\pi_\theta(o_i|q)}{\pi_{\theta_{old}}(o_i|q)}A_i, \text{clip}\left(\frac{\pi_\theta(o_i|q)}{\pi_{\theta_{old}}(o_i|q)}, 1-\epsilon, 1+\epsilon\right)A_i\right) - \beta\mathbb{D}_{KL}\left(\pi_\theta||\pi_{ref}\right)\right),$$
$$(1)$$

where $A_i$ is the advantage, computed using a group of rewards $\{r_1, r_2, \ldots, r_G\}$ corresponding to the outputs within each group:

$$A_{i,t} = \frac{r_i - \text{mean}(\{R_i\}_{i=1}^G)}{\text{std}(\{R_i\}_{i=1}^G)}. \tag{2}$$

The advantage of each response is computed as a normalized reward within a group of repeated rollouts. When all responses in a group receive the same reward, regardless of whether they are all correct or all incorrect, the resulting reward variance is zero, and the computed advantages for those responses are all zero. As a result, these examples provide no learning signal during training. In this paper, we refer to such prompts as *zero-variance prompts*, while prompts that yield non-identical rewards across responses are termed *effective prompts*.

## 3.2 Reduction of Effective Prompts in GRPO Training

The existence of zero-advantage prompts can largely reduce the effective prompt ratio in a training batch. As shown in Figure 3, during GRPO training on Qwen2.5-Math-7B [47], the ratio of effective prompts keeps decreasing as the training proceeds: at the late stage of training, this ratio can be around only 20%. A varying ratio of effective prompts can potentially hurt training stability and final model performance [49].

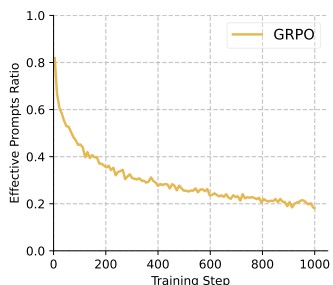

A potential way to address this instability issue is to oversample and select a batch only containing effective prompts, which is also known as Dynamic sampling (DS) [49]. As shown in Figure 2 Left, GRPO with DS consistently outperforms the vanilla GRPO, particularly on datasets such as AMC and AIME24, also with a higher average accuracy. This performance gain stems from DS's ability to filter out zero-variance prompts, thereby stabilizing training. While DS leads to better performance, it incurs significantly higher computational cost due to its need to

Figure 3: Dynamics of effective prompts ratio in each step in GRPO training. The ratio keeps decreasing as the training proceeds.

oversample more data to maintain the training batch size of effective prompts (as shown in Figure 2 Right). For instance, if the zero-variance prompt ratio is 80%, DS needs to perform around five times rollouts to maintain the training batch size. However, a substantial amount of rollout computation is wasted on prompts that ultimately result in zero-variance prompts. Identifying such prompts prior to rollout can significantly reduce computational overhead.

## 3.3 Temporal Correlation of Prompts across Epochs

Training data typically exhibits strong temporal correlations across epochs [24, 41, 55–57]. We hypothesize that zero-variance prompts in GRPO training similarly have such strong correlations in their training dynamics, enabling opportunities for more efficient identification of these prompts prior to the rollout stage. To test this hypothesis, we conduct a study on the temporal correlation of zero-variance prompts in GRPO training. Specifically, we train Qwen2.5-Math-7B with GRPO and measure two probabilities to study the temporal correlation of zero-variance prompts: **1)** P(Previous|Current): The probability that a prompt identified as zero-variance in the current epoch was also zero-variance in any previous epoch. **2)** P(Current|Previous): The probability that a prompt identified as zero-variance in any previous epoch remains zero-variance in the current epoch.

The results shown in Figure 4a indicate that zero-variance prompts exhibit strong temporal correlations throughout training. We have two key observations: *1) Prompts previously identified as zero-variance are likely to remain zero-variance.* P(Previous|Current) curve shows that the majority of zero-variance prompts in a given epoch (e.g., over 90%) were also identified as zero-variance in earlier epochs. *2) Some zero-variance prompts can become effective again in future epochs.* P(Current|Previous) curve shows that approximately 20% of prompts previously labeled as zero-variance become effective prompts that contribute to training again. This suggests that, rather than statically pruning zero-variance prompts, it is beneficial to retain some degree of exploration helps retain potentially valuable prompts.

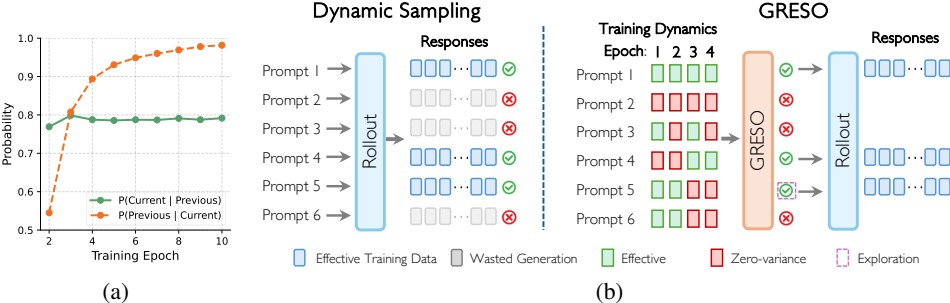

(a)                                                                (b)

Figure 4: **(a)** Temporal correlation of examples across epochs. Prompts previously identified as zero-variance are likely to remain zero-variance. **(b)** Pipeline comparison between Dynamic Sampling and our GRESO method. Unlike Dynamic Sampling, which filters out zero-variance prompts *after rollout*, GRESO efficiently predicts and filters them based on training dynamics *before rollout*, which improves rollout efficiency. The probabilistic filtering also allows zero-variance prompts to still be occasionally sampled, enabling the model to revisit potentially valuable prompts.

# 4 Methodology: GRESO

In this section, building on the two observations discussed in Section 3.2, we design GRESO (GRPO with Efficient Selective Rollout), a novel, online, efficient selective rollout algorithm that predicts and skips zero-variance prompts using reward training dynamics before the rollout stage. The overall algorithm is illustrated in Algorithm 1.

## 4.1 Detection and Filtering with Reward Training Dynamics

SOTA method [49] selects effective training data by first oversampling and then filtering out zero-variance prompts after rollout, which incurs expensive rollout overhead. Building on our observation in Section 3.3 that zero-variance prompts tend to remain uninformative in future epochs, we propose to leverage reward training dynamics to detect and filter these prompts *before rollout* to save rollout computation (as shown in Figure 4b).

More specifically, we formalize the problem of zero-variance prompt detection as follows. During training, each prompt $x_i$ is associated with a training dynamics trace:

$$T_i = (e_{i,1}, R_{i,1}), \ldots, (e_{i,n}, R_{i,n}),$$

where $e_{i,j}$ denotes the epoch number of the $j$-th sampling for example $x_i$, and $R_{i,1} = \{r_{i,1}^{(k)}\}_{k=1}^G$ represents the set of response rewards obtained in that epoch. The goal of our algorithm is to predict whether $x_i$ is a zero-variance prompt—i.e., one that yields identical rewards for all responses – based on its reward dynamics $T_i$ prior to rollout.

## 4.2 Probabilistic Pre-rollout Prompt Filtering

**Probabilistic Filtering.** To utilize the reward training dynamics, we propose a *probabilistic filtering strategy*: each prompt is calculated with a filtering probability based on its current training dynamics. As observed in Section 3.3, some zero-variance prompts can become effective again in later epochs. A key advantage of this probabilistic-based approach is that it naturally balances exploitation and exploration, allowing zero-variance prompts to still be occasionally sampled, rather than being deterministically discarded too early. This enables the model to revisit potentially valuable prompts. More specifically, given a prompt $x_i$ whose training dynamics trace is $T_i = (e_{i,1}, R_{i,1}), \ldots, (e_{i,n}, R_{i,n})$, we calculated the filtering probability by:

$$p_f(x_i) = 1 - p_e^{z_i}, \tag{3}$$

**Algorithm 1:** Training Iteration in GRESO

---

**Input:** Dataset $\mathcal{D}$; Default rollout batch size $B_r^{\text{default}}$; Training batch size $B_t$; Probability step size $\Delta p$;

Base exploration probability: $p_{easy}, p_{hard}$; Targeted zero-variance percentage: $\alpha_{easy}, \alpha_{hard}$.

1   $\mathcal{B} \leftarrow \emptyset$; $B_r \leftarrow B_r^{\text{default}}$; $n_{easy}, n_{hard}, n_{total} \leftarrow 0, 0, 0$;
   /* Rollout Stage.                                                           */

2   **repeat**

3      $\{x_i\}_{i=1}^{B_r} \leftarrow$ Sample prompts from $\mathcal{D}$ and filter with Eq. 3 until batch size $= B_r$;

4      $\{x_i, r_i\}_{i=1}^{B_r \times G} \leftarrow$ Rollout generation on $\{x_i\}_{i=1}^{B_r}$;

5      $\{x_i, r_i\}_{i=1}^{B_f \times G} \leftarrow$ filter out zero-var prompt in $\{x_i, r_i\}_{i=1}^{B_r \times G}$;

6      $n_{easy} \leftarrow n_{easy} +$ filtered easy zero-var prompt count;

7      $n_{hard} \leftarrow n_{hard} +$ filtered hard zero-var prompt count;

8      $n_{total} \leftarrow n_{total} + B_r$;

9      $\mathcal{B} \leftarrow \mathcal{B} \bigcup \{x_i, r_i\}_{i=1}^{B_f \times G}$;
   /* Adaptive rollout batch size.                                       */

10      $B_r \leftarrow \min(B_r^{\text{default}},$ Adaptive rollout batch size calculated by Eq. 6);

11 **until** $|\mathcal{B}| \geq B_t$;
   /* Adjust Base Exploration Probability.                             */

12 **if** $n_{easy}/n_{total} \geq \alpha_{easy}$ **then** $p_{easy} \leftarrow p_{easy} - \Delta p$ ;

13 **else** $p_{easy} \leftarrow p_{easy} + \Delta p$ ;

14 **if** $n_{hard}/n_{total} \geq \alpha_{hard}$ **then** $p_{hard} \leftarrow p_{hard} - \Delta p$ ;

15 **else** $p_{hard} \leftarrow p_{hard} + \Delta p$ ;
   /* GRPO Training.                                                  */

16 $\mathcal{B} \leftarrow$ select $B_t$ examples from $\mathcal{B}$;

17 Update actor model with GRPO on $\mathcal{B}$;

---

$$z_i = \max \left\{ k \in [0, n] \,\middle|\, \prod_{j=n-k+1}^{n} \mathbb{I}_{i,j} = 1 \right\}, \tag{4}$$

$$\mathbb{I}_{i,j} = \begin{cases} 1, & \text{if all rewards in } R_{i,j} \text{ are identical,} \\ 0, & \text{otherwise,} \end{cases} \tag{5}$$

where $p_e$ is the base exploration probability controlling how likely a prompt is selected for rollout. $z_i$ represents the number of most recent consecutive rollouts for prompt $x_i$ that were zero-variance.

**Self-adjustable Base Exploration Probability.** One challenge of the above probabilistic filtering algorithm lies in determining the base exploration probability, which can vary across models, datasets, and even different training stages. In addition, different base probabilities may be appropriate for easy and hard zero-variance prompts. Manually selecting the probabilities for all scenarios is impractical.

To address this challenge, GRESO employs an adaptive algorithm that automatically adjusts the base exploration probability at each training iteration (Lines 14–18 in Algorithm 1). Rather than requiring users to manually select the base probability, which can vary across different settings, GRESO only requires a target zero-variance percentage. It then automatically increases or decreases the exploration rate by a step size $\Delta_p$ based on whether the observed zero-variance percentage is above or below the target. We set $\Delta_p$ to 1% in all our evaluations. Additionally, instead of using a single base exploration probability, GRESO maintains two separate values: one for easy zero-variance prompts and another for hard ones. When computing the filtering probability $p_f(x_i)$, GRESO first determines whether $x_i$ is an easy or hard zero-variance prompt and then applies the corresponding exploration probability[3].

**Adaptive Sampling Batch Size.** In the current design of Dynamic Sampling [49], if the number of valid examples is insufficient to meet the training batch size requirement, the training performs

---

[3]We set the target zero-variance ratio to 25% for all experiments and allocate it between easy and hard prompts in an $1 : 2$ ratio (i.e., 8.3% for easy and 16.7% for hard zero-variance prompts), based on the intuition that, as models become more capable during training, more exploration on hard examples can be more beneficial. However, a more optimal allocation scheme may exist, which we leave for our future study.

rollout using a fixed batch size. However, this may result in wasted computation when only a small number of additional examples are needed to complete the training batch. To further improve rollout efficiency, GRESO adopts an adaptive rollout batch size:

$$B_{\mathrm{r}} = \min(B_{\mathrm{r}}^{\mathrm{default}}, \frac{\beta B_{\Delta}}{(1-\alpha)}), \tag{6}$$

where $B_{\mathrm{r}}^{\mathrm{default}}$ is the default rollout batch size, $B_{\Delta}$ is the number of examples needed to fill the training batch, $\alpha$ is the current zero-variance example ratio in this iteration (as some rollouts have already occurred in this iteration), and $\beta$ is a safety factor, which is fixed at $1.25$ across all our evaluations, to ensure sufficient valid examples are collected. We provide an ablation study in Section 5.3 to evaluate the contribution of this adaptive batching mechanism to GRESO's overall performance.

## 5 Experiment

In this section, we evaluate GRESO on multiple benchmarks using three different models. The evaluation results show that GRESO achieves comparable performance to Dynamic Sampling while significantly reducing rollout and training costs:

- In Section 5.2, we show that GRESO reduces up to **8M** rollouts and achieves up to **2.4×** **speedup** in rollout and **2.0× speedup** in total training time compared to Dynamic Sampling without accuracy degradation.
- In Section 5.3, we conduct a detailed study on how GRESO reduces training cost with selective rollout, and we also conduct an ablation study on the contribution of GRESO components.

### 5.1 Experimental Settings

**Models & Datasets.** We run our experiments on Qwen2.5-Math-1.5B [47], DeepSeek-R1-Distill-Qwen-1.5B [9], and Qwen2.5-Math-7B [47]. For Qwen2.5-Math-1.5B/7B models, we use $4096$ as the context length, as it is the maximum context length for those two models. For DeepSeek-R1-Distill-Qwen-1.5B, we set the context length to $8196$. For Qwen2.5-14B/32B models, we use 16k as the context length. For training datasets, we evaluate our methods on two datasets: 1) DAPO+MATH (DM): We combine the DAPO dataset [49], which contains only integer solutions, with the MATH dataset [14], which also contains LaTeX-formatted solutions. We find that training on DAPO alone can degrade performance on LaTeX-based benchmarks, so we augment it with MATH to preserve formatting diversity and improve generalization. 2) OPEN-R1 30k subset (R1): A 30,000-example subset of the OPEN-R1 math dataset [10].

**Training & Evaluation.** Our method is implemented based on verl [37] pipeline and uses vLLM [22] for rollout. We use 4xH100 for Qwen2.5-Math-1.5B training and 8xH100 for Qwen2.5-Math-7B and DeepSeek-R1-Distill-Qwen-1.5B. For benchmark datasets, we use six widely used complex mathematical reasoning benchmarks to evaluate the performance of trained models: Math500 [14, 26], AIME24 [1], AMC [2], Minerva Math [23], Gaokao [53], Olympiad Bench [13]. Similar to [43], we evaluate models on those benchmarks every 50 steps and report the performance of the checkpoint that obtains the best average performance on six benchmarks. We also include more detailed experimental settings in Appendix F.

### 5.2 End-to-end Efficiency Comparison

**No performance drop with up to 3.35× fewer rollouts.** To verify the effectiveness of GRESO, we present a comprehensive evaluation of GRESO and Dynamic Sampling (DS), which filters out zero-variance examples and resamples to fill the batch with effective data, across six math reasoning benchmarks, using three different model settings in Table 1. The models are trained on either the DAPO + MATH dataset (DM) or the Open R1 subset (OR1). We report both the performance and the number of rollouts from the checkpoint that achieves the best average performance across six benchmarks. Across all training settings, GRESO achieves comparable accuracy as DS, while significantly reducing the number of rollout samples—achieving up to 3.35× fewer rollouts. For example, on Qwen2.5-Math-7B trained on the DM dataset, GRESO achieves a comparable average accuracy to DS ($57.5\%$ vs. $57.8\%$), while reducing the number of rollouts from 13.1M to 6.3M. These results demonstrate that GRESO maintains performance while substantially lowering the cost on rollouts. Similar improvements are observed across other evaluation settings.

Table 1: Performance (%) comparison across six math reasoning benchmarks. We train three models on DAPO + MATH (DM) and the Open R1 subset (OR1). Compared to Dynamic Sampling (DS), GRESO achieves similar accuracy while significantly reducing the number of rollouts.

| Dataset | Method | Math500 | AIME24 | AMC | Gaokao | Miner. | Olymp. | Avg. | # Rollout |
|---------|--------|---------|--------|-----|--------|--------|--------|------|-----------|
| *Qwen2.5-Math-1.5B* | | | | | | | | | |
| DM | DS | 77.3 | 16.7 | 61.7 | 64.2 | 31.8 | 38.7 | 48.4 | 7.6M |
| | GRESO | 76.6 | 15.0 | 61.4 | 66.2 | 33.3 | 38.5 | 48.5 | **3.3M** |
| OR1 | DS | 77.1 | 16.7 | 50.3 | 65.5 | 30.9 | 39.7 | 46.7 | 3.8M |
| | GRESO | 76.1 | 20.0 | 50.6 | 65.1 | 30.0 | 39.2 | 46.8 | **1.6M** |
| *DeepSeek-R1-Distill-Qwen-1.5B* | | | | | | | | | |
| DM | DS | 87.9 | 36.7 | 71.7 | 78.7 | 35.3 | 55.9 | 61.0 | 2.4M |
| | GRESO | 87.7 | 36.7 | 71.1 | 78.4 | 33.9 | 55.1 | 60.5 | **1.6M** |
| OR1 | DS | 84.8 | 25.0 | 68.4 | 74.0 | 34.1 | 54.2 | 56.7 | 2.4M |
| | GRESO | 85.9 | 26.7 | 66.9 | 75.2 | 33.6 | 55.5 | 57.3 | **1.5M** |
| *Qwen2.5-Math-7B* | | | | | | | | | |
| DM | DS | 82.9 | 34.2 | 79.2 | 71.7 | 35.4 | 43.6 | 57.8 | 13.1M |
| | GRESO | 82.2 | 32.5 | 80.7 | 70.2 | 35.4 | 44.1 | 57.5 | **6.3M** |
| OR1 | DS | 82.9 | 34.2 | 63.1 | 67.3 | 34.9 | 46.3 | 54.8 | 11.4M |
| | GRESO | 82.3 | 35.0 | 64.5 | 66.8 | 36.5 | 45.7 | 55.1 | **3.4M** |

**Up to 2.4× wall-clock time speed-up in rollout and 2.0× speed-up in training.** To better understand the efficiency of our proposed methods, we report the detailed end-to-end training time (1000 steps) breakdown for different stages: rollout generation, actor model update, and other overheads (e.g., reference model and advantage calculation). Qwen2.5-Math-1.5B is trained on 4×H100 GPUs, while the other two models are trained on 8×H100 GPUs. Table 2 compares the training time breakdown between GRESO and Dynamic Sampling for models trained on the DAPO + MATH dataset. For all three models, GRESO significantly reduces rollout time—achieving up to **2.4× speedup** in rollout and **2.0× speedup** in total training time compared to DS. For instance, on Qwen2.5-Math-7B, GRESO reduces rollout time from 155.9 hours to 65.5 hours, cutting overall training time from 178.0 to 88.3 hours.

Table 2: Training time (hours) breakdown and comparison for models trained on DAPO + MATH dataset. GRESO consistently lowers rollout cost and achieves up to **2.4× speedup** in rollout and **2.0× speedup** in total training cost over Dynamic Sampling.

| Method | Training | Other | Rollout | Total |
|--------|----------|-------|---------|-------|
| *Qwen2.5-Math-1.5B* | | | | |
| DS | 8.1 | 3.6 | 41.0 (1.0×) | 52.6 (1.0×) |
| GRESO | 8.9 | 3.9 | **25.2 (1.6×)** | **37.9 (1.4×)** |
| *DeepSeek-R1-Distill-Qwen-1.5B* | | | | |
| DS | 6.1 | 3.3 | 92.4 (1.0×) | 101.9 (1.0×) |
| GRESO | 6.8 | 4.0 | **62.0 (1.5×)** | **72.7 (1.4×)** |
| *Qwen2.5-Math-7B* | | | | |
| DS | 16.1 | 6.1 | 155.9 (1.0×) | 178.0 (1.0×) |
| GRESO | 16.6 | 6.3 | **65.5 (2.4×)** | **88.3 (2.0×)** |

**Scaling to 14B and 32B models.** Besides the models presented in Table 1, we further evaluate GRESO on larger models to better verify its performance at scale. The reported accuracy represents the average performance across the six benchmarks listed in Table 1, along with AIME25. Both AIME24 and AIME25 are evaluated using avg@16 accuracy. Figure 5 compares the accuracy (%) curves of GRESO and DS on Qwen2.5-14B and Qwen2.5-32B. Both GRESO and DS exhibit similar convergence speed with respect to training steps, indicating that GRESO does not compromise model accuracy and convergence spped. However, GRESO requires

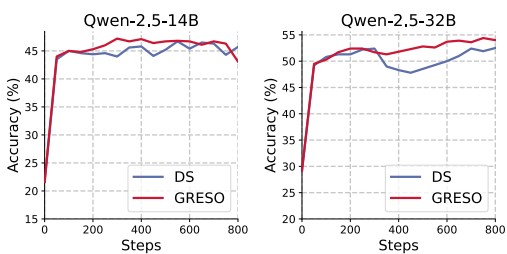

Figure 5: Accuracy (%) curves comparison between GRESO and DS on Qwen2.5-14B and Qwen2.5-32B. Both methods exhibit similar convergence speed with respect to training steps.

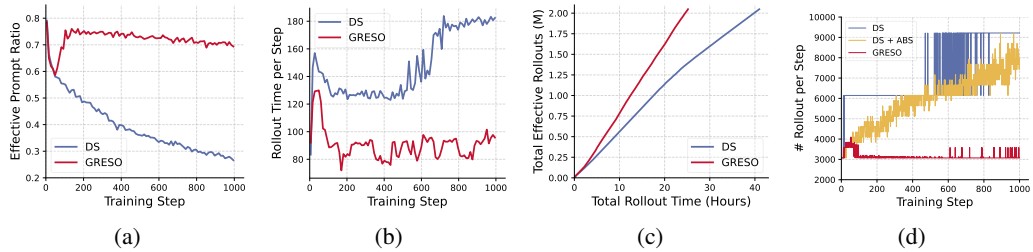

|  (a)  |  (b)  |  (c)  |  (d)  |

Figure 6: Training dynamics analysis of Qwen-Math-1.5B trained on the DAPO + MATH dataset: **(a)** Effective prompt ratio in each step. GRESO maintains a consistently higher effective prompt ratio during training. **(b)** To obtain the same number of effective prompts per batch, GRESO requires less rollout time. **(c)** GRESO achieves more effective rollouts for training under the same rollout time budget compared to Dynamic Sampling. **(d)** Ablation study on adaptive batch size (ABS) for sampling: Both ABS and GRESO effectively reduce the number of rollouts per training step.

fewer rollouts per training step. For instance, over a total of 800 training steps, Dynamic Sampling consumes 3.45M rollouts, whereas GRESO requires only 1.74M rollouts.

**Performance on coding tasks.** Besides math tasks, we also verify the effectiveness of GRESO on coding tasks. Table 3 presents a performance comparison on the `code_contests` [25] dataset using the DeepSeek-R1-Distill-Qwen-1.5B model, evaluated on `LiveCodeBench` [18]. GRESO achieves the highest accuracy, reaching 22.8%, slightly outperforming DS (22.1%) and clearly outperforming GRPO (19.9%). Besides, GRESO accomplishes this with a much lower rollout cost: 1.07M compared to 1.9M

Table 3: Performance (%) comparison on coding tasks. We train DeepSeek-R1-Distill-Qwen-1.5B on the code_contests dataset and evaluate on LiveCodeBench.

| Method | GRPO | DS | GRESO |
|---|---|---|---|
| Acc (avg@4) | 19.9 | 22.1 | **22.8** |
| # Rollouts | 0.5M | 1.9M | **1.07M** |

for DS, indicating that GRESO delivers better performance while using significantly fewer rollouts. These results demonstrate that GRESO significantly improve the *training effectiveness* without hurt model performance on code generation tasks.

**Apply GRESO on RLOO.** Besides GRPO, GRESO also enhances the efficiency of methods that suffer from the zero-variance rollout issue, like RLOO [20]. Table 4 reports the performance comparison by applying GRESO to the RLOO algorithm using the Qwen2.5-Math-7B model. Similar to applying GRESO on GRPO, when integrated with RLOO, GRESO also achieves a comparable accuracy to Dynamic Sampling (57.0% v.s. 56.8%), but need much fewer rollouts. The reported accuracy represents the average performance across the six benchmarks used in Table 1.

Table 4: Performance (%) comparison by applying GRESO on RLOO (Qwen2.5-Math-7B). Besides GRPO, GRESO also improve the efficieny of methods suffering from zero-variance rollout issue.

| Method | RLOO | RLOO+GRESO | RLOO+DS |
|---|---|---|---|
| Acc (avg@4) | 55.1 | **57.0** | 56.8 |
| # Rollouts | 0.81M | **1.2M** | 2.56M |

### 5.3 Analysis and Ablation Study

In this section, we use Qwen-Math-1.5B trained on the DAPO + MATH dataset to analyze in detail how GRESO reduces training overhead by enhancing rollout quality, and we also conduct an ablation study on the contribution of each component in GRESO.

**GRESO improves effective prompt ratio and rollout efficiency.** As shown in Figure 6a, compared to Dynamic Sampling, where effective prompt ratio steadily decreases during training, since GRESO filter out many zero-variance prompts before rollout, GRESO consistently maintains a significantly higher effective prompt ratio. For instance, as effective prompt ratio drop to around 20% in the late stage of training, GRESO maintains the effective prompt ratio larger than 70%. This higher ratio directly translates into reduced rollout time per training step, as fewer zero-variance prompts are sampled. Figure 6b shows that GRESO has significantly less rollout time per step compared to dynamic sampling. Figure 6c compares the total number of effective rollouts used during training under the same rollout time budget for GRESO and Dynamic Sampling. GRESO consistently

generates more effective rollouts over time. For instance, GRESO reaches 2 million effective rollouts in around 25 hours, while Dynamic Sampling requires over 40 hours to achieve the same, which demonstrate the efficiency of GRESO.

**Dynamics of self-adjustable base exploration probabilities.** A key parameter in GRESO is the base exploration probability $p_e$ defined in Equation 3. As discussed in Section 4.2, this probability can vary depending on the model, dataset, and training stage. Instead of manually tuning $p_e$, GRESO employs an adaptive mechanism to automatically adjust it during training. Specifically, GRESO maintains separate exploration probabilities for hard and easy zero-variance prompts, denoted as $p_{e,\text{hard}}$ and $p_{e,\text{easy}}$, respectively. In Figure 7a, we plot the dynamics of both $p_{e,\text{hard}}$ and $p_{e,\text{easy}}$, along with

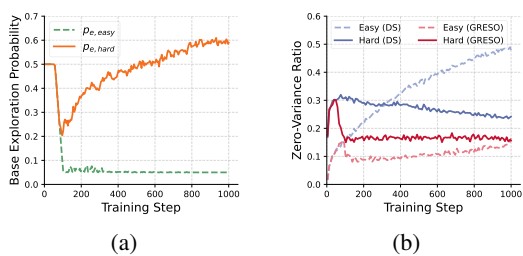

(a)                    (b)

Figure 7: **(a)** Dynamics of base exploration probabilities. **(b)** Dynamics of easy and hard zero-variance prompt ratio.

the ratio of easy and hard zero-variance prompts over time. We observe that after the first training epoch, both exploration probabilities initially decline. However, as the model ability improves, $p_{e,\text{hard}}$ begins to increase, enabling more exploration of hard examples during later stages of training. Figure 7b shows the dynamics of easy and hard zero-variance ratios. Unlike Dynamic Sampling, GRESO effectively maintains both ratios close to their target values during training, which demonstrates the effectiveness of its self-adjusting mechanism.

**Selection Dynamics.** In Figure 8, we present a case study illustrating how GRESO selects or skips prompts over training epochs. Each row stands for a prompt, and each column stands for an epoch. We observe that very easy prompts tend to remain easy throughout training; although frequently skipped, GRESO still occasionally selects them to ensure a minimal level of exploration. For prompts of moderate difficulty, as the model becomes stronger over time, these prompts gradually become easier and are increasingly skipped. In contrast, some hard prompts become solvable (i.e., effective prompts) in later epochs or even easy prompts. However, certain hard prompts remain unsolved throughout training.

**Ablation study on adaptive batch size (ABS) for sampling.** In addition to the pre-rollout prompt selection algorithm based on training dynamics, another key component of GRESO is the adaptive batch size (ABS) for sampling. When only a small number of effective prompts are needed to fill the training batch, ABS enables rollout on a smaller batch instead of using the

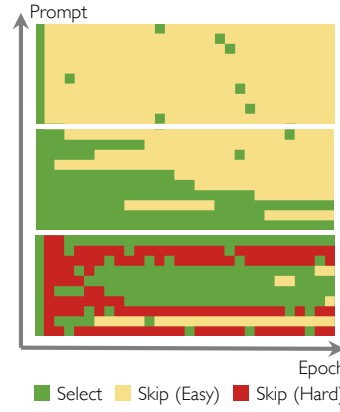

Figure 8: Selection Dynamics of different prompts in GRESO. Each row is a prompt, and each column is an epoch.

default large sampling batch size, thereby reducing unnecessary computation. Figure 6d compares the number of rollouts per training step across three methods: Dynamic Sampling (DS), DS with Adaptive Batch Size (DS + ABS), and GRESO. DS maintains a fixed sampling batch size, leading to consistently high sampling overhead. DS + ABS dynamically adjusts the batch size, reducing the number of samples in earlier steps, but still shows increasing sampling as training progresses and the effective prompt ratio decreases. In contrast, GRESO consistently maintains a much lower number of samples per step due to its more selective rollout strategy combined with ABS, resulting in significantly reduced rollout overhead.

## 6   Conclusion

In this paper, we present GRESO, a selective rollout algorithm for LLM RL. GRESO aims to improve RL training efficiency by selecting effective prompts before rollout to save unnecessary overhead on sampling uninformative prompts. GRESO leverages reward dynamics to efficiently filter out zero-variance prompts before rollout and significantly improve the RL training efficiency. Our empirical evaluation demonstrates that GRESO significantly improves end-to-end training efficiency, achieving up to $2.4\times$ rollout speedup and $2.0\times$ overall training speedup. We believe that the method and findings in our work can inspire more research on designing more efficient selective rollout algorithms to accelerate RL for LLM.

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

## Acknowledgment

LLNL-affiliated authors were supported under Contract DE-AC52-07NA27344 and supported by the LLNL-LDRD Program under Project Nos. 24-ERD-010 and 24-ERD-058. This manuscript has been authored by Lawrence Livermore National Security, LLC, under Contract No. DE-AC52-07NA27344 with the U.S. Department of Energy. The United States Government retains, and the publisher, by accepting the article for publication, acknowledges that the United States Government retains a non-exclusive, paid-up, irrevocable, world-wide license to publish or reproduce the published form of this manuscript, or allow others to do so, for United States Government purposes.

## A  Overview

We begin in Section B by discussing the limitations of our method. Section C highlights the broader societal and practical impact of improving rollout efficiency for LLM training. Section F details our experimental setup, and Section G presents additional empirical experiments and analysis.

## B  Limitations

While GRESO effectively filters out the most obvious zero-variance training prompts—those that contribute no learning signal to the model, it does not estimate or rank the value of the remaining prompts, which can also contain uninformative prompts that provide limited contribution to training. A potential future work for GRESO is to extend its filtering mechanism beyond binary decisions by incorporating a finer-grained scoring or ranking system to prioritize prompts based on their estimated training utility. Despite that, we view GRESO as an important first step toward such an advanced data selection algorithm for efficient rollout and believe it provides a solid foundation for more adaptive and efficient reinforcement learning in LLM training.

## C  Broader Impact

This work enhances the efficiency and scalability of RL-based fine-tuning for language models by introducing a lightweight, selective rollout mechanism that filters out uninformative prompts. By significantly reducing redundant computation, our method lowers overall training costs. This makes it easier for institutions with limited computational budgets to train strong models, helping democratize access to advanced AI. Furthermore, our approach promotes more sustainable and resource-efficient practices, encouraging future research toward greener and more inclusive large-scale training.

## D  Discussion

While similar temporal correlation principles have been explored in areas such as curriculum learning, how to model and leverage temporal signals to detect and filter zero-variance prompts prior to rollout remains underexplored and non-trivial. For instance, unlike prior works [4, 59] that operate on static datasets, RLVR involves online data generation, where the training data can vary significantly across iterations, making it non-trivial to apply existing approaches directly. To the best of our knowledge, our paper is the first to propose a practical and effective solution to filter out zero-variance prompts in RLVR, by tackling the following key challenges: 1) How to detect and filter uninformative prompts before performing rollouts (i.e., without access to the full training data)? 2) How to perform such detection and filtering with minimal overhead? 3) How to balance exploration and efficiency, ensuring that potentially useful prompts can be revisited in the future training stage? Each of these challenges is non-trivial in the RLVR setting. Our proposed method, GRESO, offers a lightweight mechanism that effectively addresses all three and demonstrates strong empirical performance across multiple benchmarks and RL algorithms.

## E  Reproductivity

We introduced our detailed experimental setting in Section F, and we also include our code in `https://github.com/Infini-AI-Lab/GRESO/`.

# F    Detailed Experimental Setting

**Models & Datasets.** We run our experiments on Qwen2.5-Math-1.5B [47], DeepSeek-R1-Distill-Qwen-1.5B [9], and Qwen2.5-Math-7B [47]. For Qwen2.5-Math-1.5B/7B models, we use $4096$ as the context length, as it is the maximum context length for those two models. For DeepSeek-R1-Distill-Qwen-1.5B, we set the context length to $8196$. For training datasets, we train our methods on two datasets in two settings: 1) DAPO+MATH (DM): We combine the DAPO dataset [49], which contains only integer solutions, with the MATH dataset [14], which also contains LaTeX-formatted solutions. We find that training on DAPO alone can degrade performance on LaTeX-based benchmarks, so we augment it with MATH to preserve formatting diversity and improve generalization. 2) OPEN-R1 30k subset (R1): A 30,000-example subset of the OPEN-R1 math dataset [10].

**Training.** Our method is implemented based on verl [37] pipeline and uses vLLM [22] for rollout. We use 4xH100 for Qwen2.5-Math-1.5B training and 8xH100 for Qwen2.5-Math-7B and DeepSeek-R1-Distill-Qwen-1.5B. We set the rollout temperature to $1$ for vLLM [22]. The training batch size is set to $256$, and the mini-batch size to $512$. We sample $8$ responses per prompt. We set the default rollout sampling batch size as $384$. For DeepSeek-R1-Distill-Qwen-1.5B, we set the context length to $8196$. For Qwen2.5-14B/32B and DeepSeek-R1-Distill-Qwen-1.5B for coding tasks, we set the context length to 16k. The training batch size is set to $128$, and the mini-batch size to $512$. We also sample $8$ responses per prompt. We set the default rollout sampling batch size as $192$. We train all models for $1000$ steps, and we optimize the actor model using the AdamW [29] optimizer with a constant learning rate of 1e-6. We use $\beta_1 = 0.9$, $\beta_2 = 0.999$, and apply a weight decay of $0.01$. We use the following question template to prompt the LLM. For reward assignment, we give a score of 0.1 for successfully extracting an answer and a score of 1.0 if the extracted answer is correct. Similar to [49], we remove the KL-divergence term. The optimization is performed on the parameters of the actor module wrapped with Fully Sharded Data Parallel (FSDP) [54] for efficient distributed training. We use 4 H100 for Qwen2.5-Math-1.5B training and 8 H100 for Qwen2.5-Math-7B and DeepSeek-R1-Distill-Qwen-1.5B (as it has a longer context length.) We set the targeted zero-variance percentage to $25\%$ for all experiments and allocate it between easy and hard prompts in an $1:2$ ratio (i.e., $8.3\%$ for easy and $16.7\%$ for hard zero-variance prompts), based on the intuition that, as models become more capable during training, more exploration on hard examples can be more beneficial. However, a more optimal allocation scheme may exist, which we leave for future study. We set the initial exploration probability to $50\%$ and base exploration probability adjustment step size $\Delta p$ for base exploration probability to $1\%$. We also set a minimal base exploration probability to $5\%$ to ensure a minimal level of exploration on zero-variance prompts throughout training.

**GRESO with Fixed Parameters Across All Experiments.** Although GRESO introduces a few hyperparameters, we argue that hyperparameter tuning is not a major concern in practice. We designed GRESO (e.g., self-adjustable base exploration probability) to be robust under default settings and *conducted all experiments using a single fixed set of hyperparameters across models and tasks.* The consistent performance observed across different models and tasks demonstrates that GRESO does not rely on extensive hyperparameter tuning, making it both practical and easy to integrate into existing RL fine-tuning pipelines.

**Evaluation.** For benchmark datasets, we use six widely used complex mathematical reasoning benchmarks to evaluate the performance of trained models: Math500 [14, 26], AIME24 [1], AMC [2], Minerva Math [23], Gaokao [53], Olympiad Bench [13]. Same as the training setting, For Qwen2.5-Math-1.5B/7B models, we use $4096$ as the context length. For DeepSeek-R1-Distill-Qwen-1.5B, we set the context length to $8196$. Similar to [43], we evaluate models on those benchmarks every 50 steps and report the performance of the checkpoint that obtains the best average performance on six benchmarks. We evaluate all models with temperature $=1$ and repeat the test set 4 times for evaluation stability, i.e., $pass@1(avg@4)$, for all benchmarks.

---

**Question Template**

Please solve the following math problem: {{Question Description}}. The assistant first thinks about the reasoning process step by step and then provides the user with the answer. Return the final answer in \boxed{} tags, for example \boxed{1}. Let's solve this step by step.

---

# G   Additional Experiments

## G.1   Impact of Targeted Zero-variance Percentage

In this section, we study how varying the targeted zero-variance percentage impacts training and rollout efficiency. In addition to the default setting of $25\%$ used throughout our experiments, we also evaluate alternative values of $0, 50\%, 100\%$ (i.e., always allow exploration). As shown in Table 5, different zero-variance targets give us nearly identical accuracy. We also present the number of rollouts per step in Figure 9. When we reduce the targeted zero-variance ratio to $0$, we observe that the number of rollouts per step remains similar to that of the $25\%$ setting. This lack of difference can be attributed to two factors. First, we enforce a minimum exploration rate of $5\%$, which ensures that some exploration still occurs. As a result, the actual zero-variance percentage never truly reaches $0$. Second, we always oversample some data in the first batch of rollouts in each iteration to provide some redundancy to avoid the second batch of rollouts. With this

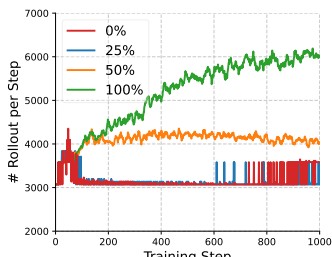

Figure 9: Comparison of the number of rollouts across different target zero-variance ratios.

setting, as long as the first batch generates enough effective training data to fill the training batch, regardless of whether the target is $0$ or $25\%$, the total number of rollouts remains approximately the same. In addition, as the targeted zero-variance percentage increases, more zero-variance prompts are allowed during rollout, leading to a higher number of rollouts per step. When the targeted percentage becomes sufficiently large, GRESO gradually approaches the behavior of dynamic sampling with adaptive rollout batch size.

Table 5: Average accuracy across six math reasoning benchmarks under different targeted zero-variance percentages.

| Target (%) | 0 | 25 | 50 | 100 |
|---|---|---|---|---|
| Acc. (%) | 48.1 | 48.5 | 48.5 | 48.4 |

## G.2   Alternative Design: Linear Backoff

In addition to the probabilistic filtering approach introduced in Section 4.2 of the main paper, we also explored an alternative solution for filtering zero-variance prompts during the early stages of this project. One such method is the *backoff algorithm* [21] (e.g., linear backoff). Specifically, if a prompt is identified as zero-variance in the most recent $k$ rollouts, it is skipped for the next $k$ training epochs. However, there are several limitations to this approach. As discussed in Section 4 of the paper, the degree of exploration should adapt to the model, dataset, and training stage. The linear backoff algorithm schedules the next rollout for a zero-variance prompt $k$ epochs into the future. As a result, if we wish to adjust the exploration intensity dynamically based on new observations or evolving training dynamics, the backoff algorithm cannot directly affect prompts that have already been deferred to future epochs. For instance, as shown in Figure 10,

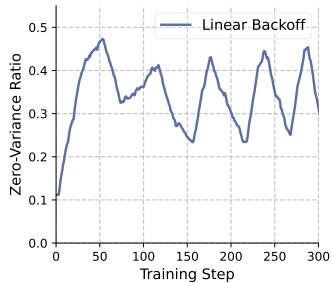

Figure 10: Zero-variance prompt ratio dynamic for linear backoff.

unlike probabilistic filtering, filtering based on linear backoff can cause periodic fluctuations in zero-variance prompt ratio, which differs from the smoother dynamics enabled by probabilistic filtering This lack of flexibility limits its ability to adapt exploration strategies in a fine-grained or responsive manner, which motivated the design of our current GRESO algorithm based on probabilistic filtering.

## G.3   Case study of Filtered Examples

To better understand the behavioral patterns of our selective filtering algorithm, we present a case study of prompts that were frequently skipped or selected during training from the MATH [14] dataset. We

categorize the examples into three groups: Frequently Skipped Prompts (Easy), Frequently Skipped Prompts (Hard), Frequently Selected Prompts. We observe that frequently skipped easy prompts often involve straightforward calculations or routine applications of formulas, making them more likely to be solved across all sampled responses. Frequently selected prompts tend to exhibit moderate difficulty, contributing more consistently to model improvement. As for frequently skipped hard prompts, these problems are too challenging for the model to solve, even across multiple rollouts, resulting in zero variance among the rewards and ultimately failing to contribute to training.

---

**Frequently Skipped Prompts (Easy)**

1. **Question:** Johnny has 7 different colored marbles in his bag. In how many ways can he choose three different marbles from his bag to play a game? **Solution:** 35.

2. **Question:** The number $n$ is a prime number between 20 and 30. If you divide $n$ by 8, the remainder is 5. What is the value of $n$? **Solution:** 29.

3. **Question:** Evaluate: $\frac{10^{-2} \cdot 5^0}{10^{-3}}$ **Solution:** 10.

4. **Question:** The Ponde family's Powerjet pumps 420 gallons of water per hour. At this rate, how many gallons of water will it pump in 45 minutes? **Solution:** 315.

5. **Question:** Suppose that $n, n+1, n+2, n+3, n+4$ are five consecutive integers. Determine a simplified expression for the sum of these five consecutive integers. **Solution:** $5n + 10$.

---

**Frequently Skipped Prompts (Hard)**

1. **Question:** A parabola and an ellipse share a focus, and the directrix of the parabola is the line containing the minor axis of the ellipse. The parabola and ellipse intersect at two points. Given that the equation of the ellipse is $\frac{x^2}{25} + \frac{y^2}{9} = 1$, find the distance between those two points. **Solution:** $\frac{4\sqrt{14}}{3}$.

2. **Question:** In triangle $ABC$, $AB = AC = 100$, and $BC = 56$. Circle $P$ has radius 16 and is tangent to $\overline{AC}$ and $\overline{BC}$. Circle $Q$ is externally tangent to $P$ and is tangent to $\overline{AB}$ and $\overline{BC}$. No point of circle $Q$ lies outside of $\triangle ABC$. The radius of circle $Q$ can be expressed in the form $m - n\sqrt{k}$, where $m$, $n$, and $k$ are positive integers and $k$ is the product of distinct primes. Find $m + nk$. **Solution:** 254.

3. **Question:** Let $EFGH$, $EFDC$, and $EHBC$ be three adjacent square faces of a cube, for which $EC = 8$, and let $A$ be the eighth vertex of the cube. Let $I$, $J$, and $K$, be the points on $\overline{EF}$, $\overline{EH}$, and $\overline{EC}$, respectively, so that $EI = EJ = EK = 2$. A solid $S$ is obtained by drilling a tunnel through the cube. The sides of the tunnel are planes parallel to $\overline{AE}$, and containing the edges $\overline{IJ}$, $\overline{JK}$, and $\overline{KI}$. The surface area of $S$, including the walls of the tunnel, is $m + n\sqrt{p}$, where $m$, $n$, and $p$ are positive integers and $p$ is not divisible by the square of any prime. Find $m + n + p$. **Solution:** 417.

4. **Question:** Let $a$ and $b$ be nonnegative real numbers such that

$$\sin(ax + b) = \sin 29x$$

for all integers $x$. Find the smallest possible value of $a$. **Solution:** $10\pi - 29$.

5. **Question:** Four people sit around a circular table, and each person will roll a standard six-sided die. What is the probability that no two people sitting next to each other will roll the same number after they each roll the die once? Express your answer as a common fraction. **Solution:** $\frac{35}{72}$.

1. **Question:** Let $x, y$, and $z$ be three positive real numbers whose sum is 1. If no one of these numbers is more than twice any other, then find the minimum value of the product $xyz$. **Solution:** $\frac{1}{32}$.

2. **Question:** The number

$$e^{7\pi i/60} + e^{17\pi i/60} + e^{27\pi i/60} + e^{37\pi i/60} + e^{47\pi i/60}$$

is expressed in the form $re^{i\theta}$, where $0 \leq \theta < 2\pi$. Find $\theta$. **Solution:** $\dfrac{9\pi}{20}$.

3. **Question:** For what values of $x$ is

$$\frac{x - 10x^2 + 25x^3}{8 - x^3}$$

nonnegative? Answer as an interval. **Solution:** $[0, 2)$.

4. **Question:** Determine all real numbers $a$ such that the inequality $|x^2 + 2ax + 3a| \leq 2$ has exactly one solution in $x$. **Solution:** 1, 2.

5. **Question:** By starting with a million and alternatively dividing by 2 and multiplying by 5, Anisha created a sequence of integers that starts 1000000, 500000, 2500000, 1250000, and so on. What is the last integer in her sequence? Express your answer in the form $a^b$, where $a$ and $b$ are positive integers and $a$ is as small as possible. **Solution:** $5^{12}$.

