# OpenReview forum: "Act Only When It Pays: Efficient Reinforcement Learning for LLM Reasoning via Selective Rollouts"
_NeurIPS.cc/2025/Conference — NeurIPS 2025 poster_

### Official Review · Reviewer_sAP6 · 2025-06-05

**Clarity:** 3
**Significance:** 2
**Originality:** 2
**Rating:** 5
**Confidence:** 4

**Summary:**

This paper introduces GRESO (GRPO with Efficient Selective Rollout), a novel method to enhance the efficiency of reinforcement learning (RL) for large language model (LLM) reasoning by selectively skipping uninformative prompts before rollout. The authors identify that prompts yielding zero advantage (zero-variance prompts) in one training epoch are likely to remain uninformative in subsequent epochs, leading to unnecessary computational overhead. GRESO leverages this temporal consistency in reward dynamics to predict and filter out such prompts proactively, using an online, lightweight algorithm with probabilistic filtering and adaptive batch sizing. Evaluated across multiple math reasoning benchmarks and models, GRESO achieves up to 2.4× speedup in rollout time and 2.0× reduction in total training time while maintaining model accuracy. The method addresses key limitations of prior approaches like Dynamic Sampling, which filters prompts after rollout, by reducing wasted computation.

**Questions:**

Besides the weaknesses above, further questions are as follows:

- Can the authors provide some insights on the exploration mechanism in GRESO?
- Since original DAPO is implemented using a 32B model, do the authors test GRESO on larger models, such as 32B?
- Can GRESO be extended to other domains beyond math reasoning, such as coding?
- How do the authors design the prompt using in training and test sets? I have tested several setups (including (1) DAPO with original prompt, (2) DAPO with DeepScaleR's prompt, (3) Skywork-OR1 with DeepScaleR's prompt and (4) Skywork-OR1 without prompt) and found that the prompt can influence on the model performance and entropy. For example, (1) has significant lower model entropy (0.62) compared with (2) (0.93) at the begining. Also, the performance of (1) drops largely after training 750 steps while (2) performs well. I would like to know the reason behind the prompt.

**Ethical Concerns:**

["NO or VERY MINOR ethics concerns only"]

**Final Justification:**

All my concerns are addressed by the author's rebuttal.

**Limitations:**

Yes.

**Paper Formatting Concerns:**

No.

**Quality:**

3

**Strengths And Weaknesses:**

#### Strengths

- The idea is novel and interesting.
- The paper is easy to follow.
- The code is provided.

#### Weaknesses

- The experiments lack baselines. Although the authors compare GRESO with DS of DAPO, the comparison is limited to a single method. A more comprehensive comparison with other methods would strengthen the paper, such as comparing with GRPO [1]. Also, I have run GRPO with DAPO+MATH dataset using DeepSeek-R1-Distill-Qwen-1.5B on 8x H800. After training 1170 steps, the model achieves 36.771 on AIME24 (avg@32) and training these steps only require 51 hours and 1.2M rollouts. Although some hyperparameters (e.g., response length) and package versions (e.g., vllm) may be different and this direct comparison is not fair, most important hyperparameters are the same by checking the provided training scripts and I suggest the authors compare GRESO with GRPO.
- The experiments lack base methods. Efficient Selective Rollout can be used with other RL algorithms such as VC-PPO [2] and VAPO [3]. Experiments with these methods would provide a more comprehensive evaluation of GRESO's effectiveness.

[1] DeepSeekMath: Pushing the Limits of Mathematical Reasoning in Open Language Models.

[2] What’s behind ppo’s collapse in long-cot? value optimization holds the secret.

[3] Vapo: Efficient and reliable reinforcement learning for advanced reasoning tasks.

---

> ### Author Rebuttal · Authors · 2025-07-31
>
> ## Response to Reviewer 4 (sAP6)
>
> Thank you very much for your thoughtful review and constructive suggestions. We are glad that you found our idea both interesting and novel, and the presentation clear. We appreciate the opportunity to address the points you raised and hope you will consider raising your score in light of our response.
>
> ---
>
> > **Q1:**  GRPO can be treated as a baseline.
>
> **A1:**  Thanks for your suggestion to compare GRESO with GRPO. We would like to address your concerns from the following two perspectives:
>
> 1. We have already included a GRPO as a baseline in Figure 1 in the main paper: GRPO without DS can achieve a worse final accuracy in some settings.
>
> 2. GRESO is complementary to GRPO and can be applied to accelerate vanilla GRPO.
>
> 3. Discussion of GRPO performance on DeepSeek-R1-Distill-Qwen-1.5B.
>
> **1. GRPO without DS can achieve a worse final accuracy in some settings.**
> We included GRPO as a baseline for training the Qwen2.5-Math-7B model in Figure 1. We show that, although GRPO training is faster, but achieves a worse final accuracy.  This phenomenon appears in several settings, like the 32B model results reported in the DAPO paper [1]. Similarly, in our additional RLOO (Q2) and Code (Q5) evaluations in the rebuttal, we also notice that vanilla GRPO achieves an inferior performance. For this reason, many recent methods adopt DS as a default component to stabilize training [1,2,3]. Therefore, in our main comparison table, we focus our comparison on DS rather than GRPO.
>
> **2.GRESO is complementary Vanilla GRPO**:  More importantly, GRESO is not incompatible with GRPO—in fact, the **GRESO and GRPO are complementary**. While our paper focuses on applying GRESO to accelerate DS, the same idea can be used to improve the efficiency of GRPO as well. Given a batch of prompts in GRPO, GRESO can be used to predict which prompts are likely to result in zero-variance rollouts. These prompts can then be skipped during rollout generation, reducing redundant computation. To verify this claim, we conduct an evaluation by applying GRESO to GRPO for training DeepSeek-R1-Distill-Qwen-1.5B on the DAPO+MATH dataset. The results are as follows:
>
> Best Accuray for AIME24 (This result exceeds the paper-reported number because Table 1 reports checkpoints selected for the best average accuracy across benchmarks, whereas here we report the best accuracy on this individual benchmark.):
>
> Best Accuray for AIME24:
>
> |AIME24|GRPO|GRPO+GRESO|
> |------|----|----------|
> |Acc(avg@4)|39.5|40.6|
> |#Rollouts|561k|322k|
>
> Best Accuray for AIME25:
>
> |AIME25|GRPO|GRPO+GRESO|
> |------|----|----------|
> |Acc(avg@4)|33.3|32.3|
> |#Rollouts|461k|377k|
>
> Total #rollouts for 1000 iterations:
>
> |1000 Iterations|GRPO|GRPO+GRESO|
> |------|----|----------|
> |#Rollouts|1.02M|650k|
>
> For both AIME24 and AIME25, GRESO achieves comparable performance while significantly reducing the number of rollouts. For all 1000 training iterations, GRPO has accumulated 1.02 million rollouts, whereas GRPO + GRESO only requires 650k—achieving over 35% reduction in rollout cost for the same number of updates. These results confirm that **GRESO not only accelerates DS training but also boosts the efficiency of vanilla GRPO by reducing redundant rollouts.**
>
>
> **3. Discussion of GRPO performance on DeepSeek-R1-Distill-Qwen-1.5B for Math.**
> Thank you for sharing the evaluation results on DeepSeek-R1-Distill-Qwen-1.5B.  We also note that the effectiveness of DS varies depending on the training dataset difficulty for the base model. For instance, when training on DeepSeek-R1-Distill-Qwen-1.5B, the observed zero-variance rate is relatively low (around 50%), allowing GRPO to maintain training quality. In contrast, models such as Qwen2.5-Math-7B exhibit much higher zero-variance rates (up to 80%) under the same setup, which significantly harms performance. Furthermore, as we will present in Q5, when training DeepSeek-R1-Distill-Qwen-1.5B on the code_contests dataset, vanilla GRPO yields inferior performance.
>
>
> We thank the reviewer again for the insightful suggestion. We will include this discussion and the additional experimental results in the revised version of the paper to better illustrate the relationship between GRESO and GRPO
>
>
> > **Q2:** Selective rollout can also be applied with other baselines like VC-PPO and VAPO.
>
> **A2:** We agree that GRESO can be applied to other RL methods. However, VC-PPO and VAPO are value-based approaches that do not suffer from the zero-variance issue in RLVR, which GRESO is specifically designed to address. To demonstrate the generality of GRESO, we applied it to RLOO [4] on the MATH + DAPO benchmark using Qwen2.5-Math-7B and report average accuracy across 6 benchmarks used in the paper. The results are as follows (we report the best accuracy and the corresponding number of rollouts at that iteration):
>
>
> |Method|RLOO|RLOO+GRESO|RLOO+DS|
> |---|---|---|----|
> |Acc(avg@4)|55.1|57|56.8|
> |#Rollouts|0.81M|1.2M|2.56M|
>
> These results show that GRESO improves accuracy over vanilla RLOO while requiring **less than half** the number of rollouts compared to RLOO+DS. This demonstrates that GRESO is both effective and efficient in other RLVR methods.
> While GRESO was not applied to value-based methods in this work, we agree that selective rollout for value-based RL is an important direction. We will include a discussion of this extension in the main paper, and we consider it a promising direction for our future research.
>
> > **Q3:**  Can the authors provide some insights on the exploration mechanism in GRESO?
>
> **A3:**  As shown by the green curve in Figure 4(a) of the main paper, approximately 20% of prompts previously labeled as zero-variance become effective again in subsequent training. This highlights the importance of maintaining a certain level of exploration—even for prompts that were previously considered uninformative.
> GRESO addresses this by computing an exploration probability for each prompt based on its recent zero-variance history. Specifically, the more frequently a prompt has been identified as zero-variance in consecutive rollouts, the less likely it is to be informative, and thus it is assigned a lower exploration probability. This adaptive mechanism allows GRESO to selectively revisit potentially useful prompts while minimizing unnecessary rollouts.
>
> > **Q4:**  Evaluation on 32B models.
>
> **A4:**  Thank you for the suggestion. We have already initiated experiments on 14B and 32B-scale models to evaluate the scalability of GRESO. Due to the high computational cost, these runs are still in progress, but we report interim results below (average accuracy across the 6 benchmarks used in the paper):
>
> |14B (150 iterations) |GRESO|DS|
> |------|----|----------|
> |Acc(avg@4)|55.4|56.3|
> |#Rollouts|353k|455k|
>
> |32B (100 iterations) |GRESO|DS|
> |------|----|----------|
> |Acc(avg@4)|60.4|59.2|
> |#Rollouts|254k|303k|
>
> These preliminary results indicate that GRESO achieves comparable accuracy to Dynamic Sampling while using fewer rollouts. We are continuing these experiments and will update the results during the discussion phase. We will also ensure that the complete results, especially for the 32B model, are included in the final version of the paper.
>
> > **Q5:**  Can GRESO be extended to other domains beyond math reasoning, such as coding?
>
> **A5:**  Yes, GRESO can be extended beyond math reasoning. To further evaluate the generalizability of GRESO, we applied it to the code generation domain. Specifically, we trained DeepSeek-R1-Distill-Qwen-1.5B on the code_contests dataset and evaluated the models on the LiveCodeBench benchmark. The results are as follows (we report the best accuracy and the corresponding number of rollouts at that iteration):
>
> |Method|GRPO|GRESO|DS|
> |----|---|-----|--|
> |Acc(avg@4)|19.9|22.8|21.6|
> |#Rollouts|0.55M|1.07M|1.8M|
>
> We trained all the models until convergence. **GRESO achieves comparable accuracy to DS with fewer rollouts**, and GRPO fails to match the final accuracy of GRESO and DS.
> These results suggest that GRESO is also **effective in other reasoning-intensive domains** such as code generation. Thank you for the suggestion, we will include the results in the final version of our paper.
>
> > **Q6:**  How are prompts designed for the paper?
>
> **A6:**  We empirically selected our prompt by adding a reasoning instruction to the default Verl prompt used for Math tasks (our final prompt is provided in Appendix E). We also evaluated the DAPO prompt during our experiments, but found that it led to inferior performance. We hypothesize that, one possible reason for the performance drop is the mismatch in answer formatting — the DeepSeek distilled model is trained to place the final answer in a box, whereas the DAPO template does not follow it.
>
> ---
>
> Thanks for the attentive reading of the manuscript and constructive feedback. We will incorporate these changes into our final version.
> We hope our response addresses all the concerns and that the reviewer will consider raising the rating accordingly. We are more than glad to answer any further questions.
>
> ---
>
> [1] DAPO: An Open-Source LLM Reinforcement Learning System at Scale
>
> [2] ProRL: Prolonged Reinforcement Learning Expands Reasoning Boundaries in Large Language Models.
>
> [3] Beyond the 80/20 Rule: High-Entropy Minority Tokens Drive Effective Reinforcement Learning for LLM Reasoning
>
> [4] BUY 4 REINFORCE SAMPLES, GET A BASELINE FOR FREE!

---

> > ### Comment · Reviewer_sAP6 · 2025-08-04
> > **Thanks for your rebuttal**
> >
> > Thanks for your detailed response. A2-A6 address my concerns. I have three follow-up questions.
> >
> > 1. The results in A1 on AIME24 and AIME25 are reported using avg@4, which is not accurate since the number of questions is merely 30. avg@16 is a more proper metric to report [1, 2]. Are the results in Table1 also reported using avg@4? I suggust the authors to provide the used evaluation metric in the paper, following [1, 2].
> >
> > 2. I have another question that when the training dataset is larger and training 1000 steps cannot reach the second epoch, how can GRESO tackle this situation? The simplest way is to training more steps but the training time will increase with more steps.
> >
> > 3. What if directly discard the questions with zero-variance advantages?
> >
> > I will increase my score if the authors address my concerns.
> >
> > References:
> >
> > [1] ProRL: Prolonged Reinforcement Learning Expands Reasoning Boundaries in Large Language Models.
> >
> > [2] Beyond the 80/20 Rule: High-Entropy Minority Tokens Drive Effective Reinforcement Learning for LLM Reasoning

---

> > > ### Author Response · Authors · 2025-08-04
> > > **Thanks for the responses and follow-up discussion.**
> > >
> > > Thanks for the responses and follow-up discussion! We are happy to address the new questions that you raised.
> > >
> > > > **Q1:** The results in A1 on AIME24 and AIME25 are reported using avg@4, which is not accurate since the number of questions is merely 30. avg@16 is a more proper metric to report [1, 2]. Are the results in Table1 also reported using avg@4? I suggust the authors to provide the used evaluation metric in the paper, following [1, 2].
> > >
> > > **A1:** We thank the reviewer for the helpful suggestion. We have updated the reported AIME24 and AIME25 results in our rebuttal to use avg@16:
> > >
> > > |AIME24|GRPO|GRPO+GRESO|
> > > |------|----|----------|
> > > |Acc(avg@16)|37.5|38.9|
> > > |#Rollouts|561k|322k|
> > >
> > > |AIME25|GRPO|GRPO+GRESO|
> > > |------|----|----------|
> > > |Acc(avg@16)|32.2|32.0|
> > > |#Rollouts|461k|377k|
> > >
> > > For the results reported in Table 1 of the main paper, we have already started re-running all the experiments to update the data, as we unfortunately did not keep all checkpoints due to space limitations. In the final version, we will make sure that both AIME24 and AIME25 are evaluated using avg@16 for a more robust evaluation.
> > >
> > > Even so, we would like to emphasize that our results are obtained from a wide range of experiments across different task types, training datasets, model scales, and benchmarks. We believe that this breadth of evaluation also serves as strong evidence for robustness and generality, demonstrating that GRESO consistently improves rollout efficiency without sacrificing performance across diverse settings.
> > >
> > > > **Q2:** I have another question that when the training dataset is larger and training 1000 steps cannot reach the second epoch, how can GRESO tackle this situation? The simplest way is to training more steps but the training time will increase with more steps.
> > >
> > > **A2:**  Multi-epoch RL training is a common practice in existing RLHF and RLVR pipelines [1,2,3,4,5]. In our experiments, we also observe that training over multiple epochs consistently leads to performance improvements. Therefore, we believe this is a valid and practical setting that can benefit from selective rollout strategies like GRESO.
> > >
> > > However, we acknowledge this as a valid limitation. One potential way to address it is to use the base model to profile the dataset before training to identify potential zero-variance examples and enable data filtering even within the first epoch. We thank the reviewer again for this insightful suggestion and will consider it as a direction for future work.
> > >
> > > > **Q3:** What if directly discard the questions with zero-variance advantages?
> > >
> > > **A3:**  In our early exploration stage, we conducted an evaluation where any example that ever appeared as a zero-variance sample was permanently discarded. However, we found that this strategy significantly degraded training performance and even caused training to collapse in some cases.
> > >
> > > There are two main reasons for this:
> > >
> > > 1. **Dropping hard but solvable questions**: Many prompts are initially too hard to solve in the early training stage. Permanently removing them prevents the model from benefiting from these harder examples in later stages.
> > >
> > >
> > >
> > > 2. **Over-filtering**: In some training settings, we observed that over 95% of the prompts may appear as zero-variance **at least once** during training. If all of these are permanently discarded, the training set rapidly shrinks, leaving too little data for effective training.
> > >
> > > These findings motivated the design of GRESO’s exploration mechanism, allowing zero-variance prompts to still be occasionally sampled.
> > >
> > > ---
> > >
> > > Thanks again for your valuable feedback and further discussion. We hope our updated responses have addressed all of your concerns. We will ensure that all new results and corresponding changes are included in the final version of the paper. Please feel free to reach out with any additional questions—we are happy to continue the conversation.
> > >
> > > ---
> > >
> > >
> > > [1] ProRL: Prolonged Reinforcement Learning Expands Reasoning Boundaries in Large Language Models.
> > >
> > >
> > > [2] Beyond the 80/20 Rule: High-Entropy Minority Tokens Drive Effective Reinforcement Learning for LLM Reasoning
> > >
> > > [3] Yu, Qiying, et al. "Dapo: An open-source llm reinforcement learning system at scale." arXiv preprint arXiv:2503.14476 (2025).
> > >
> > >
> > > [4] Yue, Yu, et al. "Vapo: Efficient and reliable reinforcement learning for advanced reasoning tasks." arXiv preprint arXiv:2504.05118 (2025).
> > >
> > >
> > > [5] Fu, Wei, et al. "AReaL: A Large-Scale Asynchronous Reinforcement Learning System for Language Reasoning." arXiv preprint arXiv:2505.24298 (2025).

---

> > > > ### Author Response · Authors · 2025-08-04
> > > > **Update on 14B and 32B models**
> > > >
> > > > **Update on 14B and 32B models.** Besides, as we are keeping training 14B and 32B models, we would like to update our latest results on 14B and 32B models, reporting the best average accuracy across the 6 benchmarks used in the paper. Consistent with our previous findings, GRESO achieves comparable or better performance while requiring significantly fewer rollouts than DS.
> > > >
> > > > |14B |GRESO|DS|
> > > > |------|----|----------|
> > > > |Avg Acc|58.2|58.3|
> > > > |#Rollouts|941k|1.66M|
> > > >
> > > > |32B | GRESO | DS |
> > > > |------|----|----------|
> > > > |Avg Acc|64.6|64.2|
> > > > |#Rollouts|768k|1.43M|
> > > >
> > > > These results further validate GRESO’s ability to improve rollout efficiency while maintaining high performance, even at large model scales.
> > > >
> > > > Note: These are intermediate results. Training is still ongoing, and we will make sure that the final version of the paper includes the complete and updated results for both 14B and 32B models.

---

> > > > > ### Comment · Reviewer_sAP6 · 2025-08-04
> > > > > **Thanks for your response**
> > > > >
> > > > > Thanks for your detailed response and all my concerned are addressed. I have increased my score to 5 and vote for acceptance.

---

### Official Review · Reviewer_vbkh · 2025-07-01

**Clarity:** 3
**Significance:** 3
**Originality:** 3
**Rating:** 4
**Confidence:** 4

**Summary:**

The paper proposes a new online data selection strategy in RL for LLM Reasoning. Based on the findings of the strong temporal consistency in prompt value: prompts that are uninformative in one epoch of training are likely to remain uninformative in near future epochs, the paper propose GRESO (GRPO with Efficient Selective Rollout), an online, lightweight pre-rollout filtering algorithm that predicts and skips uninformative prompts using reward training dynamics. By applying GRESO on models like Qwen2.5-Math-1.5B, DeepSeek-R1-Distill-Qwen-1.5B, and Qwen2.5-Math-7B, the experimental results show that GRESO achieves up to 2.4x wall-clock time speedup in rollout and up to 2.0x speedup in total training time without accuracy degradation.

**Questions:**

The proposed method seems to have many hyper parameters to tune, such as $\beta$, $\alpha$, $p_{easy}$ and $p_{hard}$, is there any principles or guidelines to tune these parameters in broader scenarios?
How is the performance and training efficiency affected by different hyperparameters?
Is the effectiveness of the algorithm robust to different hyper parameters?

**Ethical Concerns:**

["NO or VERY MINOR ethics concerns only"]

**Final Justification:**

The author's response partially resolved my concerns.
In fact, the common practice is to use an SFT/RFT model as the initial policy instead of the base model, so usually queries are not used   multiple times to avoid overfitting.

According to other review comments and their corresponding responses, I tend to keep my score.

**Limitations:**

1. The proposed method highly relies on the premise that the training queries are used multiple times in RLVR, which might not hold when the base model or initial policy is strong, which usually trains less than one epoch to avoid overfitting or reward hacking.

**Quality:**

3

**Strengths And Weaknesses:**

## Strengths
1. The method is evaluated on multiple initial policy including Qwen2.5-Math-1.5B, DeepSeek-R1-Distill-Qwen-1.5B, and Qwen2.5-Math-7B, and shows that  GRESO consistently brings significant wall-clock time speedup in rollout and speedup in total training time without accuracy degradation.
2. The analysis of training dynamics of different queries is thorough, especially the insights about the temporal consistency between uninformative queries are interesting. The statistical analysis such as Figures 5/6/7 also helps the reviewer better understands the influence of the proposed algorithm.

## Weaknesses
1. The proposed method highly relies on the premise that the training queries are used multiple times in RLVR, which might not holds when the base model is strong, which usually trains less than one epoch to avoid overfitting or reward hacking.
2. The proposed method seems to have many hyper parameters to tune, such as $\beta$, $\alpha$, $p_{easy}$ and $p_{hard}$, is there any principles or guidelines to tune these parameters in broader scenarios, and how is the performance so as the training efficiency affected by different hyper parameters?

---

> ### Author Rebuttal · Authors · 2025-07-31
>
> ## Response to Reviewer 3 (vbkh)
>
> Thank you very much for your thoughtful review and constructive suggestions. We are glad that you found our evaluation extensively shows GRESO achieves performance gain and the analysis is thorough. We appreciate the opportunity to address the points you raised and hope you will consider raising your score in light of our response.
>
> ---
>
> > **Q1:**  The proposed method highly relies on the premise that the training queries are used multiple times in RLVR, which might not holds when the base model is strong, which usually trains less than one epoch to avoid overfitting or reward hacking.
>
> **A1:** Multi-epoch RL training is a common practice in existing RLHF and RLVR pipelines [1,2,3,4], even for strong base models such as 32B-scale LLMs [1]. In our experiments, we also observe that training over multiple epochs consistently leads to performance improvements. Therefore, we believe this is a valid and practical setting that can benefit from selective rollout strategies like GRESO.
>
> However, we acknowledge this as a valid limitation. One potential way to address it is to use the base model to profile the dataset before training, enabling effective data filtering even within the first epoch. We thank the reviewer for this insightful suggestion and will work on it as a direction for future work.
>
> > **Q2:**  The proposed method seems to have many hyper parameters to tune, such as $\beta$, $\alpha$, $p_{easy}$ and $p_{hard}$, is there any principles or guidelines to tune these parameters in broader scenarios, and how is the performance so as the training efficiency affected by different hyper parameters?
>
> **A2:**  While our method introduces several hyperparameters, we argue that **these are not manually tuned for each task and do not pose a significant burden in practice**:
>
> 1. The exploration bounds $p_{\text{easy}}$ and $p_{\text{hard}}$ are automatically adjusted during training and do not require manual specification.
>
>
> 2. The target zero-variance percentage $\alpha$ is discussed in Appendix F.1. We show that GRESO is robust across a wide range of values, and as long as $\alpha$ is sufficiently high, GRESO shows pretty good performance.
>
> 3. The exploration adjustment rate $\beta$ is a parameter inherited from GRPO; we follow prior work and simply set $\beta = 0$, which works well in our setting.
>
> More importantly, we use **the same default configuration across all experiments and consistently observe efficiency gains**, indicating that GRESO works robustly without the need for extensive hyperparameter tuning.
>
> ---
>
> Thanks again for the attentive reading of the manuscript and constructive feedback. We will incorporate these changes into our final version. We hope our response addresses all the concerns and that the reviewer will consider raising the rating accordingly. We are more than glad to answer any further questions.
>
> ---
>
> [1] Yu, Qiying, et al. "Dapo: An open-source llm reinforcement learning system at scale." arXiv preprint arXiv:2503.14476 (2025).
>
> [2] Yue, Yu, et al. "Vapo: Efficient and reliable reinforcement learning for advanced reasoning tasks." arXiv preprint arXiv:2504.05118 (2025).
>
> [3] Fu, Wei, et al. "AReaL: A Large-Scale Asynchronous Reinforcement Learning System for Language Reasoning." arXiv preprint arXiv:2505.24298 (2025).
>
> [4] Wang, Shenzhi, et al. "Beyond the 80/20 rule: High-entropy minority tokens drive effective reinforcement learning for llm reasoning." arXiv preprint arXiv:2506.01939 (2025).

---

### Official Review · Reviewer_KuoT · 2025-07-01

**Clarity:** 2
**Significance:** 2
**Originality:** 2
**Rating:** 3
**Confidence:** 4

**Summary:**

This paper introduces GRESO, at selective rollout method for reinforcement learning in LLM reasoning tasks. The key observation is that prompts yielding zero reward variance (uninformative) in one training epoch are likely to remain uninformative in subsequent epochs due to temporal consistency. GRESO leverages this by predicting and skipping such prompts before rollout using reward dynamics, reducing computational overhead. The method employs probabilistic filtering with adaptive exploration rates and batch sizing. Evaluations on math reasoning tasks (using 1.5B/7B models) report up to 2.4× rollout speedup and 2.0× total training speedup compared to Dynamic Sampling, without accuracy degradation.

**Questions:**

- How does GRESO fundamentally differ from prior work on data pruning via training dynamics?

- Can GRESO scale to models >30B parameters or non-mathematical tasks (e.g., dialogue, code generation)?

**Ethical Concerns:**

["NO or VERY MINOR ethics concerns only"]

**Final Justification:**

I appreciate the authors for doing additional experiments on larger models and clarify their contribution on identifying zero-variance samples. After the discussion, I still have the following concerns:

1. The evaluation might be problematic. Note that in the rebuttal, the authors showcase the way they report the final results, which is picking up the highest performance during a time period. Here is an example given by the authors:

The test accuracy trace for the GRPO training run:

Iteration	300	350	400	450	500	550	600	650
Acc(avg@4) with (std)	17.2	18.3	18.0	19.8	19.9	19.7	19.5	19.6

Apparently, the best model is from 500 iterations (0.5M rollouts I think). The authors report the result of 19.9 with 0.55M rollouts. So I assume that the reported # rollouts corresponds to the time period, which could be longer than the time when the best model occurs. In that sense, I highly encourage the authors to limit the same number of rollouts for fair comparison. Simply picking up the best model and reporting the performance has a great uncertainty and randomness. The training curves of LLMs can differ a lot with different initial seeds and the time when the best model occurs can also differ a lot.

2. I notice that in the original version of the paper, the authors did a comprehensive evaluation over 1.5B and 7B models. I appreciate the experiments over 14B and 32B models during the rebuttal and discussion phase. But I still encourage the authors to do a comprehensive evaluuation for 14B and 32B models and put the results into Table 1 (i.e., accross six tasks).

**Limitations:**

I would encourage the authors to discuss the potential negative societal impact of their work. For example, what if the prediction is not perfect? How to solve the out-of-distribution problem?

**Quality:**

2

**Strengths And Weaknesses:**

Strengths:

- The proposed method is clearly explained and the authors attempted to address the high computational cost issue of rollouts for LLMs.

- The authors provide a practical solution with detailed implementation (adaptive batch sizing, dual exploration rates).

Weaknesses:

- The observation that zero-variance samples are not very helpful for the LLM finetuning is not novel. For example, [1] also utilized similar mechanism to filter the dataset.

- Temporal correlation of sample utility is well-known in curriculum learning and NLP tasks [2,3]. GRESO applies this concept to RL rollouts but without significant algorithmic innovation.

- This paper has a limited scope to math reasoning and small models (≤7B). Modern LLM RL (e.g., 70B+ models) that may face distinct challenges (e.g., prompt complexity, reward sparsity) are not addressed here. Speedup gains may not generalize.
There is no formal mathematical analysis of how skipping prompts affects convergence, reward shaping, or generalization beyond test accuracy.

References:

[1] Muennighoff, Niklas, et al. "s1: Simple test-time scaling." arXiv preprint arXiv:2501.19393 (2025).

[2] Zhou, Tianyi, Shengjie Wang, and Jeff Bilmes. "Curriculum learning by optimizing learning dynamics." International Conference on Artificial Intelligence and Statistics. PMLR, 2021.

[3] Bejan, Irina, Artem Sokolov, and Katja Filippova. "Make Every Example Count: On the Stability and Utility of Self-Influence for Learning from Noisy NLP Datasets." Proceedings of the 2023 Conference on Empirical Methods in Natural Language Processing. 2023.

---

> ### Author Rebuttal · Authors · 2025-07-31
>
> ## Response to Reviewer 2 (KuoT)
>
> Thank you very much for your thoughtful review and constructive suggestions. We are glad that you found our presentation clear and our solution practical, with detailed implementation. We appreciate the opportunity to address the points you raised and hope you will consider raising your score in light of our response.
>
> ---
>
> > **Q1:** The observation that zero-variance samples are not very helpful for the LLM finetuning is not novel. For example, [1] also utilized similar mechanism to filter the dataset.
>
> **A1:**  Thanks for bringing this for discussion. As acknowledged in the paper, this observation has been discussed in prior work[1,4], and we do not claim it as a contribution in our paper. Our key contribution lies in **how to identify zero-variance samples prior to performing rollouts**—a nontrivial challenge that, to the best of our knowledge, has not been addressed in prior work, and our work is the first method to provide a practical and efficient solution to this problem.
>
> Thank you for highlighting this point—we will further make this distinction clearer in the revised version.
>
> > **Q2:** Temporal correlation of sample utility is well-known in curriculum learning and NLP tasks [2,3]. GRESO applies this concept to RL rollouts but without significant algorithmic innovation.
>
> **A2:**  While we agree with the reviewer that leveraging temporal correlation in sample utility is a well-established principle in curriculum learning and active learning, we respectfully argue that **our paper tackles a fundamentally different and underexplored problem** for RLVR for LLMs, where zero-variance prompts emerge as a unique challenge, which was discussed by recent work [1,4].
>
> While similar temporal correlation principles have been explored in areas such as curriculum learning, how to model and leverage temporal signals to detect and filter zero-variance prompts prior to rollout remains underexplored and non-trivial. For instance, unlike prior works [2,3] that operate on static datasets, RLVR involves online data generation, where the training data can vary significantly across iterations, making it non-trivial to apply existing approaches directly.
>
> To the best of our knowledge, our paper is the **first** to propose a practical and effective solution to filter out zero-variance prompts in RLVR, by tackling the following key challenges:
>
> 1. How to detect and filter uninformative prompts before performing rollouts (i.e., without access to the full training data)?
>
> 2. How to perform such detection and filtering with minimal overhead?
>
> 3. How to balance exploration and efficiency, ensuring that potentially useful prompts can be revisited in the future training stage?
>
> Each of these challenges is non-trivial in the RLVR setting. Our proposed method, GRESO, offers a lightweight mechanism that effectively addresses all three and demonstrates strong empirical performance across multiple benchmarks and RL algorithms.
>
> However, we agree with the reviewer that a discussion of these works would strengthen the paper. We will conduct a comprehensive discussion of the related prior works, including [2,3], on curriculum learning and active learning in the revised version to better clarify the connections and differences.
>
>
> > **Q3:** This paper has a limited scope to math reasoning and small models (≤7B). Modern LLM RL (e.g., 70B+ models) that may face distinct challenges (e.g., prompt complexity, reward sparsity) are not addressed here. Speedup gains may not generalize. There is no formal mathematical analysis of how skipping prompts affects convergence, reward shaping, or generalization beyond test accuracy.
>
> **A3:**
>
> **Training on Code Tasks**: To further evaluate the generalizability of GRESO, we applied it to the code generation domain. Specifically, we trained DeepSeek-R1-Distill-Qwen-1.5B on the code_contests dataset and evaluated the models on the LiveCodeBench benchmark. The results are as follows (we report the best accuracy and the corresponding number of rollouts at that iteration):
>
> |Method|GRPO|GRESO|DS|
> |----|---|-----|--|
> |Acc(avg@4)|19.9|22.8|21.6|
> |#Rollouts|0.55M|1.07M|1.8M|
>
> We trained all the models until convergence. **GRESO achieves comparable accuracy to DS with fewer rollouts**, and GRPO fails to match the final accuracy of GRESO and DS.
>
> These results suggest that GRESO is also **effective in other reasoning domains** such as code generation.
>
> **Scale to 32B model.** Thank you for the suggestion. We have already initiated experiments on 14B and 32B-scale models to evaluate the scalability of GRESO. Due to the high computational cost, these runs are still in progress, but we report interim results below (average accuracy across the 6 benchmarks used in the paper):
>
> |14B (150 iterations) |GRESO|DS|
> |------|----|----------|
> |Acc(avg@4)|55.4|56.3|
> |#Rollouts|353k|455k|
>
> |32B (100 iterations) |GRESO|DS|
> |------|----|----------|
> |Acc(avg@4)|60.4|59.2|
> |#Rollouts|254k|303k|
>
> These preliminary results indicate that GRESO achieves comparable accuracy to Dynamic Sampling while using fewer rollouts. We are continuing these experiments and will update the results during the discussion phase. We will also ensure that the complete results, especially for the 32B model, are included in the final version of the paper.
>
> Besides, we agree that a formal theoretical understanding of the mechanism of our method is an important and open-ended research question. The primary goal of this work is to develop a practical and effective method for improving rollout efficiency. GRESO demonstrates consistent speed-up across multiple models and domains, which we believe provides strong evidence of its effectiveness in practice. We hope that our method and findings can serve as a foundation for future theoretical investigations into the dynamics of selective rollout in RL-based LLM training.
>
>
> ---
>
> Thanks for the attentive reading of the manuscript and constructive feedback. We will incorporate these changes into our final version.
> We hope our response addresses all the concerns and that the reviewer will consider raising the rating accordingly. We are more than glad to answer any further questions.
>
> ---
>
> [1] Muennighoff, Niklas, et al. "s1: Simple test-time scaling." arXiv preprint arXiv:2501.19393 (2025).
>
> [2] Zhou, Tianyi, Shengjie Wang, and Jeff Bilmes. "Curriculum learning by optimizing learning dynamics." International Conference on Artificial Intelligence and Statistics. PMLR, 2021.
>
> [3] Bejan, Irina, Artem Sokolov, and Katja Filippova. "Make Every Example Count: On the Stability and Utility of Self-Influence for Learning from Noisy NLP Datasets." Proceedings of the 2023 Conference on Empirical Methods in Natural Language Processing. 2023.
>
> [4] Yu, Qiying, et al. "Dapo: An open-source llm reinforcement learning system at scale." arXiv preprint arXiv:2503.14476 (2025).

---

> ### Comment · Reviewer_KuoT · 2025-08-01
>
> Thank you for your response. I still have the following concerns:
>
> - In the leaderboard of the LiveCodeBench benchmark, it utlizes pass@1 as the metric. Could you please also report the performance comparison with this metric?
>
> - From your first table, it is difficult to infer that your proposed method is better than GRPO since we cannot tell if the GRPO algorithm has converged or not.
>
> - Additionally, if collecting more rollouts (i.e., with the same number of rollouts as DS), will your algorithm outperform DS when the model is 14B/32B?
>
> - It is better to include mean and std for the reported results.

---

> > ### Author Response · Authors · 2025-08-01
> > **Thanks for the responses and follow-up discussion.**
> >
> > Thanks for the responses and follow-up discussion! We are happy to address the new concerns that you raised.
> >
> > > **Q1:** In the leaderboard of the LiveCodeBench benchmark, it utlizes pass@1 as the metric. Could you please also report the performance comparison with this metric?
> >
> > **A1:**  Thank you for the question. We would like to clarify that the Avg@4 metric we report is actually pass@1, but averaged over four independent runs (i.e., we compute pass@1 across four different eval runs and report the average). This evaluation protocol is consistent with the leaderboard setting on LiveCodeBench, and thus our reported results are directly comparable.
> >
> > > **Q2:** From your first table, it is difficult to infer that your proposed method is better than GRPO since we cannot tell if the GRPO algorithm has converged or not.
> >
> > **A2:** In the first table (code reasoning), we ensured that all methods—including GRPO, GRESO, and DS—were trained for sufficient iterations to allow convergence, defined as the point after which no further test accuracy improvement is observed (typically after 100–200 iterations).
> > For instance, below is the test accuracy trace from the GRPO training run:
> >
> >
> > For instance, here is the test accuracy trace for the GRPO training run:
> > | Iteration | 300  | 350  | 400  | 450  | 500  | 550  | 600  | 650  |
> > |-----------|------|------|------|------|------|------|------|------|
> > | Acc(avg@4) with (std)  | 17.2 | 18.3  | 18.0 | 19.8 | **19.9** | 19.7 | 19.5 | 19.6 |
> >
> > As shown above, the accuracy plateaus after around 500 iterations, which we consider an indication of convergence. Therefore, we believe the reported results offer a fair and representative comparison.
> >
> > > **Q3:** Additionally, if collecting more rollouts (i.e., with the same number of rollouts as DS), will your algorithm outperform DS when the model is 14B/32B?
> >
> > **A3:** Yes, as shown in our earlier response, the current results on 32B already indicate that GRESO outperforms DS in accuracy while using fewer rollouts.
> >
> > While these results are intermediate, we already observe a **positive trend**—GRESO achieves a better trade-off between rollout efficiency and model accuracy. We appreciate the reviewer’s insightful question and will ensure that the **final version of the paper includes comprehensive and finalized results for both 14B and 32B models**.
> >
> > We also want to note that these large-scale experiments are computationally expensive and run on many nodes, which are occasionally preempted. As a result, some runs may be delayed. Nevertheless, we are continuing the experiments and will update additional results during the discussion phase.
> >
> >
> > > **Q4:** It is better to include mean and std for the reported results.
> >
> > **A4:**  Thanks for the suggestions! The reported accuracy results already reflect the mean over multiple runs (or repeated sampling). We have now added the standard deviation (std) to provide a clearer picture of the performance variability for the three tables in the previous response. These updates will be included in the revised version of the paper.
> >
> > Evaluation of code reasoning:
> >
> > |Method|GRPO|GRESO|DS|
> > |----|---|-----|--|
> > |Acc(avg@4) with (std) |19.9 (0.37) |22.8 (0.4) |21.6 (0.47)|
> > |#Rollouts|0.55M|1.07M|1.8M|
> >
> >
> > 14B and 32B evaluation of math reasoning. The standard deviation is the average standard deviation across 6 benchmarks:
> >
> > |14B (150 iterations) |GRESO|DS|
> > |------|----|----------|
> > |Acc(avg@4) with (std) |55.4 (0.31) |56.3 (0.15)|
> > |#Rollouts|353k|455k|
> >
> > |32B (100 iterations) |GRESO|DS|
> > |------|----|----------|
> > |Acc(avg@4) with (std) |60.4 (0.26) | 59.2 (0.14)|
> > |#Rollouts|254k|303k|
> >
> > ---
> >
> > Thanks again for your valuable feedback and further discussion. We hope our updated responses have addressed all of your concerns. We will ensure that all new results and corresponding changes are included in the final version of the paper. Please feel free to reach out with any additional questions—we are happy to continue the conversation.

---

> > > ### Author Response · Authors · 2025-08-04
> > > **Update on 14B and 32B models**
> > >
> > > Dear Reviewer 2 (KuoT),
> > >
> > > Thank you again for your thoughtful feedback and insightful discussion. We believe your comments have been instrumental in helping us improve the quality of the paper. As we continue our large-scale training runs, we would like to share updated results on the 14B and 32B models, which we hope further address your concerns.
> > >
> > > **Update on 14B and 32B models.** We would like to update our latest results on 14B and 32B models, reporting the best average accuracy across the 6 benchmarks used in the paper. Consistent with our previous findings, GRESO achieves comparable or better performance while requiring significantly fewer rollouts than DS.
> > >
> > > |14B |GRESO|DS|
> > > |------|----|----------|
> > > |Avg Acc|58.2|58.3|
> > > |#Rollouts|941k|1.66M|
> > >
> > > |32B | GRESO | DS |
> > > |------|----|----------|
> > > |Avg Acc|64.6|64.2|
> > > |#Rollouts|768k|1.43M|
> > >
> > > These results further validate GRESO’s ability to improve rollout efficiency while maintaining comparable performance, even at large model scales.
> > >
> > > Note: These are intermediate results. Training is still ongoing, and we will make sure that the final version of the paper includes the complete and updated results for both 14B and 32B models.
> > >
> > > ---
> > >
> > > Thanks again for your valuable feedback and further discussion. We hope our updated responses have addressed all of your concerns. We will ensure that all new results and corresponding changes are included in the final version of the paper. Please feel free to reach out with any additional questions—we are happy to continue the conversation.

---

> > > > ### Comment · Reviewer_KuoT · 2025-08-09
> > > >
> > > > Thank you for your updated results. It would be great to include a comprehensive evaluation of 14B/32B models in your next version. And it is highly encouraged to append the training curves (evidence of convergence) to the appendix to make the results more convincing.

---

### Official Review · Reviewer_3M88 · 2025-07-06

**Clarity:** 3
**Significance:** 2
**Originality:** 3
**Rating:** 4
**Confidence:** 4

**Summary:**

This paper introduces GRESO, a lightweight rollout filtering strategy for efficient reinforcement learning (RL) on large language models (LLMs). GRESO dynamically adjusts each prompt’s sampling probability to limit the frequency of zero-variance rollouts and thereby conserve consumption, all with few hyperparameters to tune. On mathematical reasoning tasks, it achieves substantial reductions in rollout cost without performance loss. Its key innovation lies in an automatic control of sampling probability to maintain a preset zero-variance target by monitoring the sampling history, ensuring both exploration and efficiency.

**Questions:**

Please see weaknesses.

**Ethical Concerns:**

["NO or VERY MINOR ethics concerns only"]

**Limitations:**

Yes

**Quality:**

3

**Strengths And Weaknesses:**

Strengths：

1. The paper addresses an important challenge of rollout efficiency in RL on LLMs with a simple yet effective approach.
2. The design of the dynamic control of exploration probability is insightful and well motivated.
3. The manuscript is well writen and easy to follow, and the purpose of each component is convincingly justified.

Weaknesses：

1. The experiments on DeepSeek-R1-Distill-Qwen-1.5B seems problematic: an 8192 context length is clearly too short and likely incurs a high truncation rate during training. The authors should either include a more robust setting in the main result or explain their choice.
2. Although the zero-variance target is chosen intuitively and leave the discussion to future work, a thorough ablation on this parameter would be highly informative and would significantly strengthen the paper.
3. The ablation study of adaptive batch size is not entirely convincing. If the comparison is based on rollouts per step, then the optimal batch size would be 1 per GPU, which is obviously unreasonable. A direct comparison of actual rollout time savings with ABS should be provided.
4. Several issues in the empirical results need further clarification:
4.1 In Figure 6, the zero-variance ratio of easy problems fails to converge to the target, and the corresponding base exploration probability is bounded excessively high. This may result from too large a Δp; if so, the method should be revised to handle this case.
4.2 In Figure 5(d), the rollout per step for GRESO shows large spikes after step 600, which seems abnormal given the continuous adaptation of batch size.
4.3 The DS/GRESO rollout ratio of DM-Qwen2.5-1.5B is smaller than in Figure 5 (a). Does this indicate a performance drop when training steps are fixed? Including a step-to-accuracy curve would help address this concern.

---

> ### Author Rebuttal · Authors · 2025-07-31
>
> ## Response to Reviewer 1 (3M88)
>
> Thank you very much for your thoughtful review and constructive suggestions. We are glad that the reviewer found our work addresses an important challenge in RL for LLMs, and the dynamic control of exploration probability is insightful. We appreciate the opportunity to address the points you have raised.
>
> > **Q1:**  The context of Deepseek model is 8k, which can introduce a high truncation rate.
>
> **A1:**  Thank you for pointing this out. We are currently running the 32k training and will make sure to put the results in the final version. We also note that, despite using truncation, we use the same setting for baselines and our method to guarantee the **fairness of the comparison** in the evaluation.
>
> In the meantime, we would like to respectfully argue that such a truncation setting is also **commonly used in the previous works [1,2,3,4]** to improve training efficiency, as longer context lengths significantly increase memory usage and wall-clock time.
>
> Importantly, we view the context length constraints can be treated as a special reward constraint to encourage models to reason in a limited context length (e.g., 8k). During training, we observed that the initial truncation rate was around 50%, but the model quickly adapted—dropping to below 3% after approximately 200 iterations—while continuing to improve in accuracy.
>
> We appreciate the reviewer’s concern and will include a more detailed discussion in the final version. We will also include the 32k results in the final version.
>
> > **Q2:**  An ablation study on zero-variance target can strengthen the paper.
>
> **A2:** Thank you for the suggestion. We have conducted an ablation study on the targeted zero-variance percentage, as presented in Appendix F.1, showing consistent improvement across a wide range of zero-variance targets. Our results also show that when the percentage is sufficiently high (e.g., 25% used in our experiments), GRESO matches the final accuracy of dynamic sampling. Importantly, **all our RL training experiments adopt the same target value**, indicating that this parameter is non-sensitive and easy to select in practice.
>
> > **Q3:**  Ablation study of adaptive batch size should also consider actual rollout time savings.
>
> **A3:**  We appreciate the reviewer’s insightful observation. It is indeed true that using a batch size of 1 would minimize the number of rollouts per step, but not the overall rollout time. This is because inference latency for LLMs scales nonlinearly with batch size—small batch sizes are not time-efficient due to under-utilized resources, while large batch sizes can significantly reduce per-sample cost.
>
> To illustrate this, we conducted an experiment using Qwen2.5-Math-1.5B on 4 H100 to measure the actual rollout time at different batch sizes. The results are shown below:
>
> | batch size | 384   | 256  | 192  | 128  | 64   | 32  | 16  |
> |---|---|---|---|---|---|---|---|
> | time   | 135.2| 94.4 | 70   | 57.48| 43   | 36  | 35  |
>
> We observe that when the batch size is small (e.g., ≤64), increasing the batch size yields per-sample rollout little time saving. However, beyond a certain point, the rollout time increases almost linearly with batch size. Therefore, by using ABS to reduce the batch size in this regime, we can significantly reduce overall rollout time without sacrificing efficiency.
>
> In our training setup, we set the default rollout batch size to 384. When ABS reduces the batch size adaptively (e.g., from 384 to 128), it leads to a substantial reduction in actual rollout time. This is the core motivation behind ABS, and we will clarify this point in the final version of the paper.
>
> > **Q4.1:**  Figure 6: the zero-variance ratio of easy problems fails to converge to the target, and the corresponding base exploration probability is bounded excessively high. This may result from too large a Δp; if so, the method should be revised to handle this case.
>
> **A4.1:**  Thank you for pointing this out. The reason the base exploration probability appears excessively high is that we enforce a minimum exploration rate (5%) to ensure a non-zero level of exploration. This lower bound helps prevent the model from entirely ignoring potentially informative examples. While this constraint may prevent the zero-variance ratio from fully converging to the target, it guarantees a certain level of exploration throughout training. Importantly, even without fully reaching the target ratio, our method still eliminates a significant amount of unnecessary rollouts and achieves substantial speedup. We will revise the manuscript to clarify this design choice and avoid potential confusion.
>
> > **Q4.2:** In Figure 5(d), the rollout per step for GRESO shows large spikes after step 600, which seems abnormal given the continuous adaptation of batch size.
>
> **A4.2:** In the rollout stage, we enforce a minimum rollout batch size of 64, as small-batch LLM inference does not yield significant differences in wall-clock time (as discussed in Q3). The spikes observed after step 600 in Figure 5(d) are caused by GRESO triggering a second round of rollouts, during which an additional 64 × 8 responses are sampled. We will clarify this behavior in the revised version to avoid confusion.
>
> > **Q4.3:** The DS/GRESO rollout ratio of DM-Qwen2.5-1.5B is smaller than in Figure 5 (a). Does this indicate a performance drop when training steps are fixed? Including a step-to-accuracy curve would help address this concern.
>
> **A4.3:** The rollout ratio in Figure 5(a) reflects the per-step ratio, while the cumulative ratio is much closer to the values reported in Table 1. In Table 5, we report the best-performing checkpoints during training, evaluated every 50 iterations as described in Appendix D. We observed that the final iteration is not always the best one. However, after convergence, the accuracy tends to fluctuate slightly rather than showing a clear performance drop. We will include a step-to-accuracy curve in the revised version of our paper.
>
> ---
>
> Thanks again for the attentive reading of the manuscript and constructive feedback. We will incorporate these changes into our final version.
> We hope our response addresses all the concerns and that the reviewer will consider raising the rating accordingly. We are more than glad to answer any further questions!
>
> ---
>
> [1] Wang, Yiping, et al. "Reinforcement learning for reasoning in large language models with one training example." arXiv preprint arXiv:2504.20571 (2025).
>
> [2] Liu, Mingjie, et al. "Prorl: Prolonged reinforcement learning expands reasoning boundaries in large language models." arXiv preprint arXiv:2505.24864 (2025).
>
> [3] Cui, Ganqu, et al. "The entropy mechanism of reinforcement learning for reasoning language models." arXiv preprint arXiv:2505.22617 (2025).
>
> [4] Cheng, Daixuan, et al. "Reasoning with exploration: An entropy perspective." arXiv preprint arXiv:2506.14758 (2025).

---

> > ### Author Response · Authors · 2025-08-06
> > **Update on 32k context length**
> >
> > **Update on 32k context.** We have updated our results on  DeepSeek-R1-Distill-Qwen-1.5B with 32k context length, reporting the best average accuracy across the 6 benchmarks used in the paper. Consistent with our previous findings, GRESO achieves comparable performance while requiring significantly fewer rollouts than DS.
> >
> > |32k |GRESO|DS|
> > |------|----|----------|
> > |Avg Acc|62.4|62.3|
> > |#Rollouts|419k|706k|
> >
> > These results further support GRESO’s effectiveness in improving rollout efficiency while maintaining comparable performance.
> >
> > ---
> >
> > Thanks again for your valuable feedback and further discussion. We hope our updated responses have addressed your concerns. We will ensure that all new results and corresponding changes are included in the final version of the paper. Please feel free to reach out with any additional questions—we are happy to continue the conversation.

---

### Note · Authors · 2025-08-14

Dear AC and all reviewers,

We sincerely thank the AC and all reviewers for their constructive reviews and helpful discussions throughout the review process. These reviews and discussions have greatly enhanced the quality and clarity of the work. **We will make sure to include those new results and discussion in our final version of the paper**.


**1. [R1, R2, R4] Additional evaluations:** In response to reviewers, we have added extensive new evaluations, including **large-scale (14B/32B) models, long-context (32k) settings, coding benchmarks, an additional baseline, and ablation studies and analysis.** Across all evaluations, GRESO consistently achieves comparable accuracy while requiring significantly fewer rollouts, demonstrating its efficiency. The new ablation results further validate our design choices, showing that each component of GRESO meaningfully contributes to its performance gains. Reviewer 4 (sAP6), who initially raised similar evaluation concerns, acknowledged these improvements and increased the rating from 3 to 5.


**2. [R2] Novelty and significance:** We clarified to Reviewer 2 that zero-variance prompts are a unique challenge in RLVR, arising from its online, dynamic data generation, and cannot be addressed by previous static-dataset methods. To the best of our knowledge, **our paper is the first to propose a practical and effective solution to detect and filter out zero-variance prompts before rollouts**, thereby accelerating RLVR training without hurting accuracy. This contribution addresses an urgent and underexplored problem and lays the groundwork for future work in efficient RL for LLM reasoning.


**3. [R1, R2, R3, R4] Methodology and results clarity:**  We have refined our methodology descriptions to clearly explain the motivation, algorithmic design, and integration into the RLVR pipeline. We also clarified the training setup, evaluation protocols, and interpretation of results to ensure they can be easily understood and reproduced.

---

Overall, we believe the additional evaluations, detailed clarifications, and further discussion have addressed the reviewers’ major concerns. We will ensure that all new results, evaluations, and clarifications are included in the final version of the paper.

---

Thanks again,

Authors of Act Only When It Pays: Efficient Reinforcement Learning for LLM Reasoning via Selective Rollouts

---

### Decision · Program_Chairs · 2025-09-17

**Decision:**

Accept (poster)

**Comment:**

This paper proposes GRESO, a lightweight pre-rollout filtering algorithm that predicts and skips uninformative prompts in reinforcement learning for LLM reasoning, achieving up to 2.4x rollout speedup without accuracy degradation.

The paper addresses an important computational bottleneck in LLM RL training with a practical solution. The authors demonstrated strong responsiveness to reviewer feedback, providing extensive additional experiments across model scales (up to 32B) and domains (math and coding). The adaptive filtering mechanism represents a solid technical contribution. The method's demonstrated effectiveness across diverse settings and substantial computational savings make it a valuable contribution to the field.

Given the efforts the authors made during the rebuttal process, most reviewers provided positive reviews and found the approach insightful and well-motivated with good experimental design. Reviewer KuoT strongly suggested running comprehensive evaluations of large models and including training curves in the paper. Given the limited resources available for academic research and the extensive experiments already conducted on 1.5B/7B models with up to 32k context length, this remaining issue does not constitute a solid reason to reject this paper. However, I encourage the authors to follow the reviewer's suggestions and update the manuscript in the camera-ready version, which will definitely improve the solidness of the original draft.

**Overall, I recommend accepting this paper.**